# Cerebral small vessel disease genomics and its implications across the lifespan

Muralidharan Sargurupremraj et al.[#]

White matter hyperintensities (WMH) are the most common brain-imaging feature of cerebral small vessel disease (SVD), hypertension being the main known risk factor. Here, we identify 27 genome-wide loci for WMH-volume in a cohort of 50,970 older individuals, accounting for modification/confounding by hypertension. Aggregated WMH risk variants were associated with altered white matter integrity (p = 2.5×10-7) in brain images from 1,738 young healthy adults, providing insight into the lifetime impact of SVD genetic risk. Mendelian randomization suggested causal association of increasing WMH-volume with stroke, Alzheimer-type dementia, and of increasing blood pressure (BP) with larger WMH-volume, notably also in persons without clinical hypertension. Transcriptome-wide colocalization analyses showed association of WMH-volume with expression of 39 genes, of which four encode known drug targets. Finally, we provide insight into BP-independent biological pathways underlying SVD and suggest potential for genetic stratification of high-risk individuals and for genetically-informed prioritization of drug targets for prevention trials.

[#]A list of authors and their affiliations appears at the end of the paper.

As a leading cause of stroke, cognitive decline, and dementia, cerebral small vessel disease (SVD) represents a major source of morbidity and mortality in aging populations[1–3]. Exploring the mechanisms of SVD and their contribution to dementia risk has recently been identified as a priority research area[4,5], based on its more frequent recognition with magnetic resonance imaging (MRI), its high prevalence in older community persons[3,6] and the demonstration that intensive management of vascular risk factors, especially hypertension, may slow down its progression[7,8]. The biological underpinnings of SVD are poorly understood and no mechanism-based treatments currently are available. White matter hyperintensities of presumed vascular origin (WMH), the most common MRI-marker of SVD, can be measured quantitatively using automated software. They are highly heritable[9], and confer an increased risk of stroke and dementia[3], thus making them well-suited to identify potential genetic determinants of SVD and its contribution to stroke and dementia risk. WMH are most often covert, i.e., not associated with a history of clinical stroke. They are highly prevalent in the general population, and much more frequently observed than clinical stroke caused by SVD (which can be both ischemic [small vessel stroke] and hemorrhagic [deep intracerebral hemorrhage]) (Supplementary Fig. 1).

Studying the genomics of SVD also provides a powerful approach to discovery of underlying molecular mechanisms and targets in order to accelerate the development of future therapies, or identify drug repositioning opportunities[10–12]. Although genomic studies of WMH have been most fruitful for deciphering SVD risk variants compared with other MRI-features of SVD (lacunes, cerebral microbleeds, dilated perivascular spaces)[13] or small vessel stroke[14], or deep intracerebral hemorrhage[15], few risk loci have been identified to date[16–18]. This is likely due to limited sample size of populations studied and possibly also the failure to take into account the role of hypertension (HTN), the strongest known risk factor for WMH, in confounding or modifying genetic associations. There is also mounting evidence suggesting that early-life factors play a crucial role in the occurrence of late-life vascular and neurological conditions, including SVD[19], likely due to both genetic and environmental factors that may intrinsically influence the vascular substrate of SVD or modulate the brain's resilience to SVD[20–22]. Identifying these early predictors could have major implications for our understanding of disease mechanisms across the lifespan and for devising effective prevention strategies.

Here, we conduct a large multiancestry meta-analysis of WMH-volume genome-wide association studies (GWAS), accounting for HTN as a potential confounder and effect modifier. We explore association of WMH risk loci with early changes in white matter microstructure on MRI using diffusion tensor imaging (DTI) in young adults in their early twenties. Last, we explore biological pathways underlying the observed genetic associations with SVD and their clinical significance through shared genetic variation and Mendelian randomization (MR) experiments with vascular risk factors and neurological traits, linking them with multiple epigenomic, transcriptomic, and drug-target databases.

## Results

**Genetic discovery from association analyses.** Figure 1 summarizes the overall workflow of our study that included data from 50,970 participants (N = 48,454 Europeans and 2516 African-Americans) from population-based studies taking part in the Cohorts for Heart and Aging Research in Genomic Epidemiology (CHARGE)[23] consortium and from the UK Biobank (Supplementary Data 1). The mean age of participants was $66.0 \pm 7.5$

years, 53% were women and 52% hypertensive (Methods, Supplementary Methods 1, and Supplementary Data 1 for cohort-specific characteristics). There was no evidence for systematic inflation of SNP-WMH association statistics at the individual cohort or meta-analysis level (Supplementary Data 3 and Supplementary Fig. 2) for the three types of analyses performed (Methods).

In the European-only SNP-main-effects analysis, 22 independent loci harbored common variants associated with WMH volume at genome-wide significance ($P < 5 \times 10^{-8}$, Table 1, Fig. 2), lead SNPs for each independent locus were confirmed by both LD clumping and GCTA-COJO[24]. Additionally, the *NID2* locus reached genome-wide significance by the joint effect of multiple SNPs ($P = 4.87 \times 10^{-8}$, Supplementary Data 4), with $P = 5.45 \times 10^{-8}$ for lead SNP rs72680374, using GCTA-COJO (Methods, Supplementary Fig. 3). The African–American-only analysis identified a genome-wide significant locus at *ECHDC3* (Supplementary Data 4). For loci showing heterogeneity in allelic effects across ancestry groups (*PHet* < 0.01), using MR-MEGA[25] the *ECHDC3* locus and another locus near KCNK2 reached genome-wide significance (Table 1, Supplementary Data 5). In the HTN-adjusted model two additional loci were associated with WMH volume at $P < 5 \times 10^{-8}$ (*PKN2* and *XKR6*), while three loci were no longer genome-wide significant (Table 1). The 2-df genome-wide gene-HTN interaction joint meta-analysis (JMA) did not identify any additional locus (Table 1, Supplementary Data 6). Five loci reached genome-wide significance in the small African–American-only JMA, but these were not maintained in the fixed-effects multiancestry JMA (Supplementary Data 7).

In total, 27 loci reached genome-wide significance in association with WMH volume in at least one of the aforementioned analyses, of which 18 have not previously been reported (Table 1, Fig. 2). Associations with WMH volume at these loci were similar in participants with and without HTN and when stratifying on quartiles of genetically predicted SBP and DBP levels (Methods, Supplementary Data 8-10). In aggregate, however, a weighted genetic risk score of independent WMH risk loci (WMH wGRS, Methods) showed significant 1-df interaction with HTN in association with WMH volume ($\beta_{GRSxHTN} = 0.15$, $P_{GRSxHTN} = 0.009$, Supplementary Fig. 4). One previously described risk locus for WMH burden did not reach genome-wide significance in the current analysis (near *PMF1*, $P = 3.9 \times 10^{-4}$). Of note one genome-wide significant locus (*COL4A2*) and one suggestive locus (*HTRA1*, $P < 5 \times 10^{-6}$, Supplementary Data 11), involve genes implicated in monogenic forms of SVD[26,27].

Additional, gene-based tests using MAGMA[28] yielded 49 gene-wide significant associations ($P < 2.8 \times 10^{-6}$), of which 13 were outside GWAS loci, including the *APOE* gene (Methods, Supplementary Data 12). Using the Heritability Estimator from Summary Statistics (HESS)[29] we found that $29 \pm 2\%$ of WMH-volume variance is explained by common and low frequency variants across the genome, the amount of heritability attributable to loci containing GWAS index SNPs being $2.4 \pm 0.1\%$.

**Implications of WMH genes across the lifespan.** To examine the lifetime impact of WMH risk variants on brain structure, we explored the association of the WMH wGRS with MRI-markers of white matter microstructure in 1738 young healthy adults participating in the i-Share cohort (mean age $22.1 \pm 2.3$ years, 72% women). Integrity of the white matter microstructure was measured on diffusion tensor imaging (DTI) using the following metrics: fractional anisotropy (FA), mean diffusivity (MD), radial diffusivity (RD), axial diffusivity ($A_xD$), and the recently described peak width of skeletonized mean diffusivity (PSMD)[30]. These MRI-markers are associated with the maturation and aging

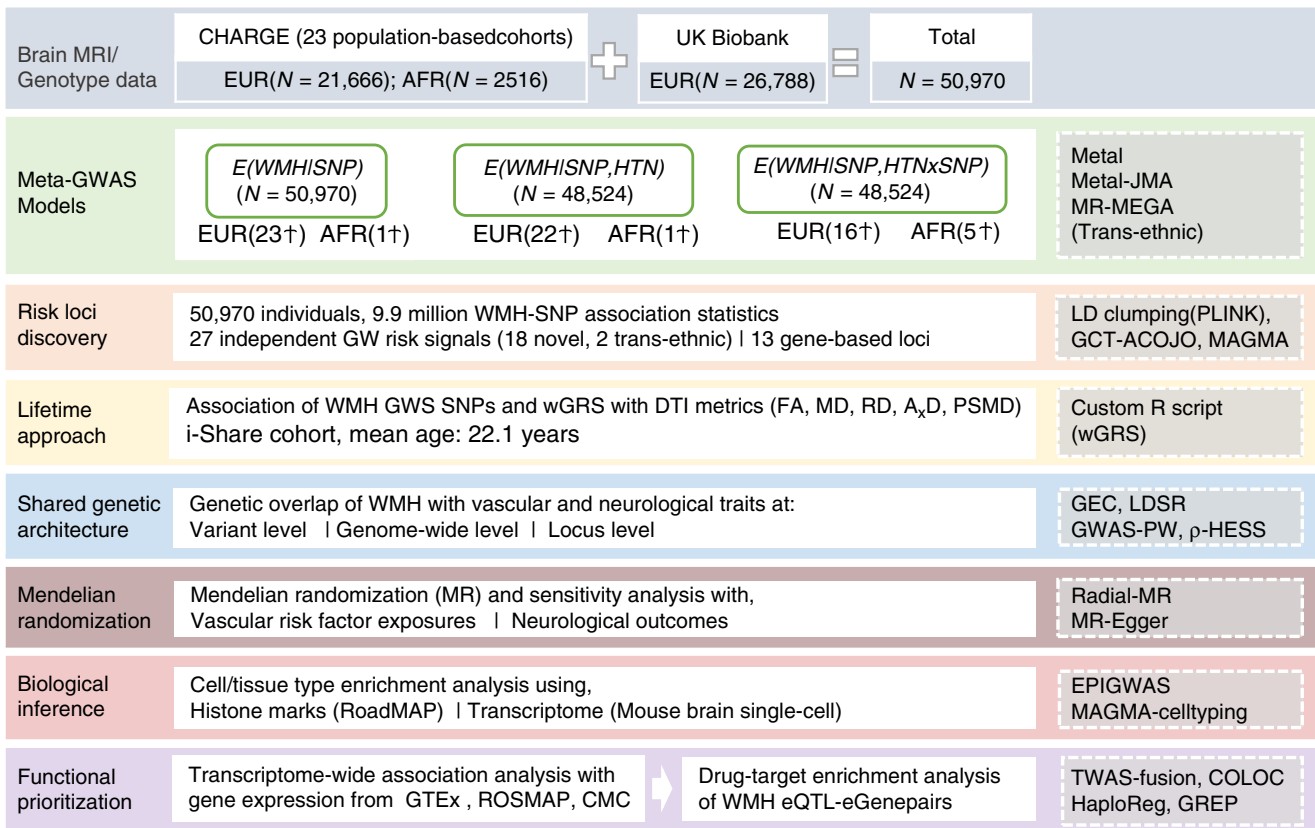

**Fig. 1 Study workflow and rationale.** † number of GW hits. MRI magnetic resonance imaging, CHARGE cohorts for heart and aging research in genomic epidemiology, EUR European, AFR African–american, GWAS genome-wide association study, WMH White matter hyperintensities, SNP single nucleotide polymorphism, HTN hypertension, JMA joint meta-analysis, MR-MEGA meta-regression of multi-ethnic genetic association, GW genome-wide, LD linkage disequilibrium, GCTA-COJO genome-wide complex trait analysis- conditional and joint analysis, MAGMA multi-marker analysis of genomic annotation, DTI diffusion tensor imaging, iSHARE internet based student health research enterprise, FA fractional anisotropy, MD mean diffusivity, RD radial diffusivity, AxD axial diffusivity, PSMD peak width of skeletonised mean diffusivity, wGRS weighted genetic risk score, GEC genetic type I error calculator, LDSR LD-score regression, GWAS-PW GWAS-pairwise analysis, HESS heritability estimator from summary statistics, EPIGWAS epigenome wide association study, TWAS transcriptome-wide association study, GTEx genotype-tissue expression, ROSMAP religious orders study and the RUSH memory and aging project, CMC common mind consortium, eQTL expression quantitative trait loci, eGene expression-associated genes, COLOC colocalisation, GREP genome for repositioning drugs.

process of white matter microstructure[31,32], and alterations in DTI metrics were shown to precede the occurrence of WMH lesions in older patients with SVD[30,33]. The WMH wGRS showed significant associations with higher MD, RD, and PSMD and lower FA values in i-Share; four WMH risk loci individually showed significant associations with at least one of the DTI parameters (*SH3PXD2A*, *NMT1*, *KLHL24*, and *VCAN*, Table 2). Increasing values of PSMD (but not other DTI markers) shows a trend towards association with slower information processing speed on the Stroop test in i-Share participants ($N = 1,401$, effect estimate ± SE: $0.085 \pm 0.040$, $P = 0.031$), which did not survive correction for multiple testing (for three independent DTI markers). The WMH wGRS was not associated with the Stroop test in i-Share but showed a trend towards an association with poorer episodic memory performance in older community persons (N = 24,597, effect estimate ± SE: -0.19 ± 0.11, $P = 0.08$)[34].

We also examined whether genetically predicted larger WMH volume was associated with increased risk of stroke and Alzheimer-type dementia, the most common age-related neurological diseases, and with lower cognitive performance in older age, using previously reported GWAS data (Supplementary Table 2). Several genome-wide significant WMH risk loci showed significant association with ischemic stroke (three loci), all stroke and small vessel stroke (two loci each), cardioembolic stroke, deep intracerebral hemorrhage,

and Alzheimer-type dementia (one locus each) (Supplementary Data 13). Using linkage disequilibrium score regression (LDSR)[35], we observed significant genetic correlation of WMH volume with all stroke, ischemic stroke, small vessel stroke, and lower general cognitive function, after Bonferroni correction for multiple testing ($P < 3.6 \times 10^{-3}$, Methods, Fig. 3, Supplementary Data 14). Using the Bayesian pairwise GWAS (GWAS-PW) approach[36], significant regional overlap (posterior probability of association for model 3, PPA3 ≥ 0.90, Methods) was observed between WMH volume and general cognitive function and between WMH volume and stroke, especially ischemic and small vessel stroke (Supplementary Data 15). This included regions previously implicated in complex and monogenic forms of stroke (*FOXF2/FOXQ1*[37,38], *HTRA1*[27,37]) and cardiovascular disease (*NOS3*)[39].

Using two-sample MR, which implements the inverse-variance weighting (IVW) method, we observed evidence for significant causal associations after Bonferroni correction for multiple testing ($P < 3.6 \times 10^{-3}$) between WMH volume and increased risk of Alzheimer-type dementia, with no statistical evidence of horizontal pleiotropy using Cochran's Q statistic (Q-PHet≥0.01, Fig. 4, Supplementary Data 16). We also observed evidence for significant causal association of WMH volume with risk of any stroke, ischemic stroke, small vessel stroke, and deep intracerebral hemorrhage. There was some evidence for horizontal pleiotropy

**Table 1 Loci reaching genome-wide significance with WMH burden in one or more genetic association model.**

| Region | Position | Nearest gene | Analysis | SNP | Local $h^2$ | Function | EA/OA | EAF | Main effects[a] | | | | | | JMA (N = 48,524) |
| | | | | | | | | | Model 1 (N = 50,970) | | | Model 2 (N = 48,524) | | | |
| | | | | | | | | | β | SE | P | β | SE | P | P |
| *Novel* | | | | | | | | | | | | | | | |
| 17q21.31 | 43144218 | NMT1[§] | EUR/TRANS | rs6503417 | 0.23% | Intronic | c/t | 0.63 | 0.052 | 0.006 | 3.43E-19 | 0.054 | 0.006 | 4.85E-20 | 9.17E-20 |
| 16q24.2 | 87227397 | C16orf95 | EUR/TRANS | rs12921170 | 0.17% | Intergenic | a/g | 0.58 | 0.050 | 0.006 | 9.82E-18 | 0.049 | 0.006 | 4.91E-17 | 1.44E-16 |
| 3q27.1 | 183363263 | KLHL24 | EUR/TRANS | rs6797002 | 0.12% | Intronic | c/t | 0.73 | 0.049 | 0.007 | 8.18E-14 | 0.046 | 0.007 | 3.26E-12 | 3.12E-11 |
| 5q14.2 | 82859065 | VCAN | EUR/TRANS | rs17205972 | 0.07% | Intronic | t/g | 0.20 | 0.049 | 0.007 | 4.76E-12 | 0.051 | 0.007 | 2.34E-12 | 4.19E-12 |
| 2q32.1 | 188028317 | CALCRL | EUR/TRANS | rs62172472 | 0.10% | Intergenic | g/a | 0.79 | 0.047 | 0.007 | 3.67E-11 | 0.046 | 0.007 | 1.73E-10 | 2.24E-10 |
| 5q23.2 | 121518378 | LOC100505841 | EUR/TRANS | rs2303655 | 0.03% | Downstream | t/c | 0.78 | 0.048 | 0.007 | 4.03E-11 | 0.049 | 0.007 | 3.58E-11 | 1.28E-11 |
| 16q12.1 | 51442679 | SALL1 | EUR/TRANS | rs1948948 | 0.02% | Intergenic | c/t | 0.56 | 0.037 | 0.006 | 1.17E-10 | 0.037 | 0.006 | 2.06E-10 | 7.27E-10 |
| 14q32.11 | 91884655 | CCDC88C | EUR/TRANS | rs1285847 | 0.10% | Upstream | t/c | 0.55 | 0.036 | 0.006 | 1.24E-10 | 0.039 | 0.006 | 1.30E-11 | 9.00E-11 |
| 8p23.1 | 8179639 | SGK223 | EUR/TRANS | rs7184312 | 0.07% | Intronic | g/a | 0.72 | 0.038 | 0.006 | 2.18E-09 | 0.035 | 0.006 | 4.51E-08 | 8.12E-08 |
| 10q24.33 | 105507145 | SH3PXD2A-AS1 | EUR/TRANS | rs71471298 | 0.21% | Intronic | t/c | 0.11 | 0.053 | 0.009 | 2.72E-09 | 0.051 | 0.009 | 2.91E-08 | 4.58E-08 |
| 1q41 | 215137222 | KCNK2[b] | TRANS | rs6540873 | 0.03% | Intergenic | a/c | 0.61 | 0.027 | 0.007 | 1.37E-08 | 0.027 | 0.007 | 2.81E-08 | 4.31E-05 |
| 10p14 | 11804452 | ECHDC3[c] | AFR/TRANS | rs1125731l | 0.01% | Intronic | g/t | 0.70 | 0.046 | 0.010 | 2.01E-08 | 0.186 | 0.032 | 4.29E-09 | 1.85E-02 |
| 22q12.1 | 27887471 | MN1 | EUR | rs5762197 | 0.05% | Intergenic | c/a | 0.71 | 0.039 | 0.007 | 2.67E-08 | 0.040 | 0.007 | 2.26E-08 | 1.01E-07 |
| 15q22.31 | 65355468 | RASL12 | EUR | rs12443113 | 0.07% | Intronic | g/a | 0.55 | 0.031 | 0.006 | 3.42E-08 | 0.029 | 0.006 | 3.43E-07 | 2.14E-06 |
| 8p23.1 | 9628753 | TNKS | EUR | rs11249945 | 0.11% | Intronic | a/g | 0.35 | 0.034 | 0.006 | 3.60E-08 | 0.032 | 0.006 | 3.84E-07 | 9.58E-08 |
| 14q22.1 | 52604843 | NID2[d] | EUR | rs72680374 | 0.07% | Intergenic | t/a | 0.63 | 0.032 | 0.006 | 5.45E-08 | 0.032 | 0.006 | 1.43E-07 | 2.42E-06 |
| 8p23.1 | 11031472 | XKR6 | EUR | rs7004825 | 0.08% | Intronic | t/c | 0.47 | 0.031 | 0.006 | 9.18E-08 | 0.032 | 0.006 | 4.59E-08 | 4.55E-08 |
| 1p22.2 | 89286673 | PKN2 | EUR | rs786921 | 0.05% | Intronic | a/g | 0.60 | 0.033 | 0.007 | 2.52E-07 | 0.037 | 0.007 | 2.97E-08 | 1.74E-05 |
| *Previously described* | | | | | | | | | | | | | | | |
| 17q25.1 | 73888354 | TRIM65 | EUR/TRANS | rs34974290 | 0.40% | Exonic | a/g | 0.19 | 0.104 | 0.007 | 2.59E-46 | 0.104 | 0.007 | 1.27E-45 | 1.34E-45 |
| 2p16.1 | 56128091 | EFEMP1 | EUR/TRANS | rs7596872 | 0.09% | Intronic | a/c | 0.10 | 0.097 | 0.010 | 3.81E-24 | 0.097 | 0.010 | 2.45E-23 | 8.06E-22 |
| 6q25.1 | 151018909 | PLEKHG1 | EUR/TRANS | rs6940540 | 0.07% | Intronic | g/t | 0.41 | 0.045 | 0.006 | 7.17E-15 | 0.043 | 0.006 | 1.32E-13 | 1.70E-13 |
| 2p21 | 43132224 | HAAO | EUR/TRANS | rs73923006 | 0.05% | Intergenic | g/c | 0.81 | 0.055 | 0.007 | 2.00E-14 | 0.053 | 0.007 | 4.28E-13 | 3.57E-12 |
| 10q24.33 | 105459916 | SH3PXD2A | EUR/TRANS | rs4630202 | 0.21% | Intronic | g/a | 0.71 | 0.048 | 0.007 | 1.46E-13 | 0.048 | 0.007 | 2.00E-13 | 3.10E-13 |
| 2q33.2 | 203780515 | CARF1 | EUR/TRANS | rs7603972 | 0.14% | Intronic | a/g | 0.87 | 0.070 | 0.010 | 2.23E-13 | 0.069 | 0.010 | 1.55E-12 | 2.80E-11 |
| 10q24.33 | 105610326 | SH3PXD2A | EUR/TRANS | rs10786772 | 0.21% | Intronic | g/a | 0.64 | 0.042 | 0.006 | 1.56E-12 | 0.042 | 0.006 | 2.26E-12 | 4.80E-11 |
| 13q34 | 111043309 | COL4A2 | EUR/TRANS | rs55940034 | 0.04% | Intronic | g/a | 0.29 | 0.037 | 0.006 | 4.49E-09 | 0.037 | 0.006 | 4.63E-09 | 2.68E-08 |
| 14q32.2 | 100625902 | DEGS2 | EUR/TRANS | rs7157599 | 0.01% | Exonic | c/t | 0.29 | 0.041 | 0.007 | 1.49E-08 | 0.042 | 0.007 | 6.76E-09 | 2.38E-06 |

For each locus, the variant reaching the lowest P value in the fixed-effects transancestral meta-analysis or the fixed-effects Europeans-only meta-analysis, respectively, is shown.
JMA joint meta-analysis, SNP single nucleotide polymorphism, EUR European, AFR African-American, TRANS transethnic, SNPs GW significant in both the analysis, association statistic from the TRANS analysis are shown, Local $h^2$ local SNP heritability for the locus estimated from SNP-main-effects EUR only, EA effect allele, OA other allele, EAF effect allele frequency, β effect estimate, SE standard error, P P value.
[a]Main effects are assessed in Model 1, adjusted for age, sex, principal components for population stratification, and total intracranial volume, and in Model 2, which is additionally adjusted for hypertension status.
[b]Locus reaching GW significance in MR-MEGA meta-regression analysis; § the lead SNP is not in LD ($r^2 < 0.01$) with the chr17q21 locus previously reported to be associated with WMH volume in stroke patients[16].
[c]Locus reaching GW significance in AFR-only and MR-MEGA meta-regression analysis.
[d]Additional locus reaching GW significance in the GCTA-COJO analysis for the main-effects model.

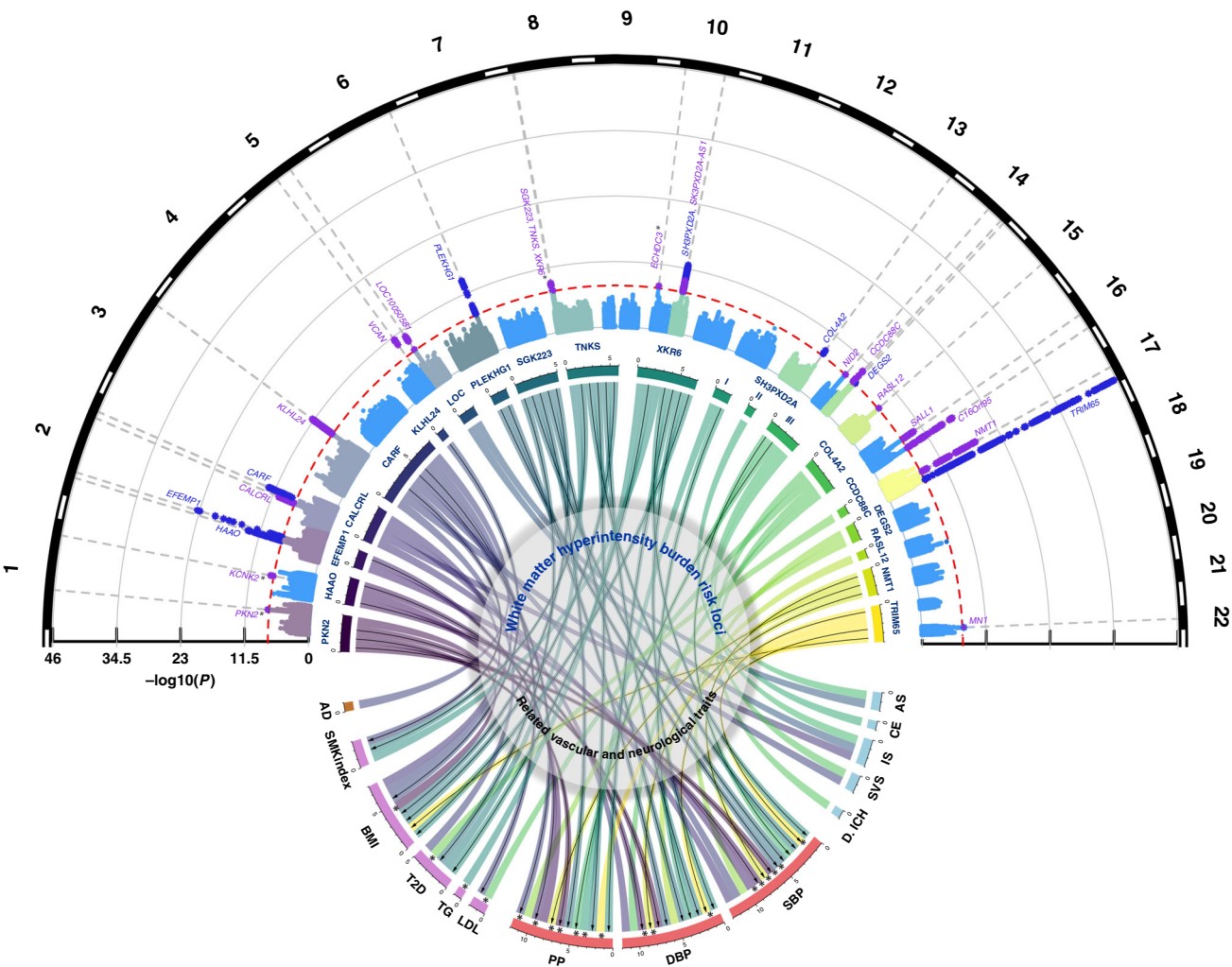

**Fig. 2 Genome-wide association results with WMH burden and genetic overlap of WMH risk loci.** Circular Manhattan plot (top) displaying novel (violet) and known (dark blue) genome-wide significant WMH risk loci (dotted line: $P < 5 \times 10^{-8}$). Asterisks denote association signals that reach genome-wide significance only in the HTN-adjusted model (*PKN2, XKR6*) or in the MR-MEGA transethnic meta-analysis (*KCNK2, ECHDC3*). Chord diagram (center) summarizing the association of genome-wide significant risk variants for WMH burden (upper section) with vascular and neurological traits (bottom section) ($P < 1.3 \times 10^{-4}$, see Methods). The width of each of the stems corresponds to the number of traits associated with a given locus (upper section) or the number of loci associated with a given trait. Black arrows indicate genome-wide significant associations, and asterisks denote SNPs exhibiting unexpected directionality of associations (WMH risk allele displaying protective association with vascular or neurological traits). LOC* LOC100505841, SBP systolic blood pressure, DBP diastolic blood pressure, PP pulse pressure, AS all stroke, IS ischemic stroke, SVS small vessel stroke, CE cardioembolic stroke, ICH intracerebral hemorrhage, AD Alzheimer's disease, BMI body mass index, LDL low-density lipoprotein, VTE venous thromboembolism, T2D type II diabetes, MIG migraine, SMKindex lifetime smoking index, WMH white matter hyperintensity.

(Q-PHet<0.01) for associations with stroke. However, after removing influential outlier variants associations remained significant and the MR-Egger intercept (indicating average pleiotropic effects) did not significantly differ from zero with $Q_R$ values close to 1, indicating goodness of fit of the IVW method (Methods, Supplementary Data 16)[40].

**Shared genetic risk with vascular traits.** To assess whether genetic associations with WMH reflect known vascular mechanisms we systematically explored the shared genetic variation between WMH burden and related vascular traits, comprising established risk factors for vascular disease (systolic blood pressure [SBP], diastolic blood pressure [DBP], pulse pressure [PP], type 2 diabetes [T2D], low-density lipoprotein [LDL] cholesterol, high-density lipoprotein [HDL] cholesterol, triglycerides [TG], body mass index [BMI], glycated hemoglobin levels

[HbA1c], and lifetime smoking index—a composite measure capturing smoking heaviness and duration as well as smoking initiation [SMKindex]), as well as putative risk factors for other disorders including venous thromboembolism (VTE) and migraine, using summary statistics from the most recent GWAS. Some of the latter were made available through collaborations with the relevant research consortia when the data were not publicly available (Supplementary Table 2).

First, we looked up associations of the 27 WMH risk loci with related vascular traits, including the lead WMH risk variants and nearby variants (±250 kb) in moderate to high LD ($r^2 > 0.5$). After correcting for the number of independent loci and traits tested ($P < 1.3 \times 10^{-4}$, Methods), 20 of the 27 WMH risk loci (74%) showed significant association with at least one other trait and/or vascular risk factors. For 13 of these, associations were found at a genome-wide significant level (Fig. 2, Supplementary Data 13). Blood pressure (BP) traits showed by far the largest number of

**Table 2 Association trend of WMH risk loci with white matter microstructures in young adults.**

| Variables | Region | Nearest gene | PSMD ($10^{-4}$ mm²s⁻¹) | | | FA | | | MD ($10^{-4}$ mm²s⁻¹) | | | RD ($10^{-4}$ m²s⁻¹) | | | $A_xD$ ($10^{-4}$ mm²s⁻¹) | | |
|---|---|---|---|---|---|---|---|---|---|---|---|---|---|---|---|---|---|
| | | | B | SE | P | β | SE | P | β | SE | P | β | SE | P | β | SE | P |
| wGRS | | | 0.077 | 0.022 | 4.53E-04 | −0.012 | 0.002 | 2.53E-07 | 0.111 | 0.026 | 1.66E-05 | 0.142 | 0.028 | 5.38E-07 | 0.050 | 0.031 | 1.09E-01 |
| SNPs[a] | | | | | | | | | | | | | | | | | |
| rs10786772 | 10q24.33 | SH3PXD2A | 0.019 | 0.005 | 9.98E-05 | −0.002 | 0.001 | 3.70E-04 | 0.022 | 0.006 | 1.11E-04 | 0.025 | 0.006 | 7.16E-05 | 0.017 | 0.007 | 1.86E-02 |
| rs6503417 | 17q21.31 | NMT1 | 0.015 | 0.005 | 2.25E-03 | −0.002 | 0.001 | 2.20E-03 | 0.021 | 0.006 | 1.52E-04 | 0.023 | 0.006 | 2.13E-04 | 0.018 | 0.007 | 7.06E-03 |
| rs830179 | 3q27.1 | KLHL24 | 0.012 | 0.005 | 1.52E-02 | −0.001 | 0.001 | 8.81E-02 | 0.014 | 0.006 | 1.32E-02 | 0.014 | 0.006 | 2.56E-02 | 0.015 | 0.007 | 2.94E-02 |
| rs17205972 | 5q14.2 | VCAN | 0.030 | 0.006 | 3.28E-07 | −0.004 | 0.001 | 2.29E-09 | 0.041 | 0.007 | 1.80E-09 | 0.049 | 0.008 | 1.20E-10 | 0.027 | 0.008 | 1.23E-03 |

All models are adjusted for age, sex, the first four principal components for population stratification, and total intracranial volume.
PSMD peak width of skeletonized mean diffusivity, FA fractional anisotropy, MD mean diffusivity, RD radial diffusivity, $A_xD$ axial diffusivity, β effect estimate, SE standard error, P P value, wGRS weighted Genetic Risk Score, SNP single nucleotide polymorphism.
[a]For individual SNPs, a p value < 2 × 10⁻³ is considered significant, after multiple testing correction (considering 25 independent loci tested).

significant associations with WMH risk variants, 16 loci (59.3%) being associated with SBP, DBP, and/or PP. Further significant associations with WMH risk variants were observed for BMI (8 loci), T2D (5 loci), SMKindex (3 loci), and lipid traits (3 loci), one locus (at *XKR6*) being notably shared with all these risk factors. Seven loci (*C16orf95, ECHDC3, MN1, NID2, SALL1, VCAN, KCNK2*), none of which were reported previously as WMH risk loci, appear not to be associated with any of the vascular traits explored, suggesting other underlying biological pathways.

Second, we explored the genome-wide and regional genetic overlap between WMH volume and related vascular traits. Mean $X^2$ ranged between 1.06 and 3.99 suggesting strong polygenicity for all investigated traits. The impact of possible sample overlap was estimated to be negligible using LDSR[35] (Supplementary Data 14). We observed significant ($P < 3.6 \times 10^{-3}$) genetic correlation of larger WMH volume with higher SBP, DBP, SMKindex, BMI and increased risk of VTE. Using GWAS-PW[36] and HESS[41] (Methods), we identified 16 genomic regions harboring shared genetic risk variants with at least one other vascular trait, predominantly BP traits, but also BMI, lipid levels and SMKindex (PPA3 ≥ 0.90, Fig. 3, Supplementary Data 15).

Third, we explored the causal relations between the aforementioned vascular traits and WMH volume using two-sample MR[42] (RadialMR[40], Methods), implementing the IVW method. We observed significant ($P < 3.6 \times 10^{-3}$) association of genetically predicted SBP, DBP, PP, SMKindex and T2D with larger WMH volume and of genetically predicted migraine with smaller WMH volume (Fig. 4, Supplementary Data 17). After removal of potentially pleiotropic outlier variants; for SBP, DBP, PP and SMKindex the MR-Egger intercept was nonsignificant, indicating no residual pleiotropy and suggesting causal association with WMH volume (Methods). For migraine and T2D in contrast there was evidence of residual pleiotropic effects (significant MR-Egger intercept, Supplementary Data 17) after removal of potentially pleiotropic outlier variants, and the association became only nominally significant for migraine. Importantly, associations of genetically predicted SBP and DBP with WMH volume remained significant after adjustment for HTN, and in participants with and without HTN (Supplementary Data 17), highlighting that higher levels of BP are likely causally associated with larger WMH volume even below BP thresholds typically used for the definition of hypertension (SBP ≥ 140 mmHg or DBP ≥ 90 mmHg or antihypertensive drug intake)[43].

**Biological interpretation of association signals**. We used EPIGWAS[44] to test for cell-type enrichment of WMH association signals using chromatin marks previously shown to be cell-type specific and associated with active gene-regulation (Methods). WMH risk loci were significantly enriched in enhancer (H3K4me1) and promoter sites (H3K4me3) in cell-types derived from the brain, neurosphere (developing brain), vascular tissue, digestive, epithelial and muscle tissues, as well as human embryonic stem cells after removing WMH risk loci associated with BP (Supplementary Data 19). Analysis of brain-specific single-cell expression data in mice using MAGMA.celltyping[45] (Methods) revealed significant enrichment of highly cell-type-specific genes in endothelial mural cells and nominally significant enrichment for vascular leptomeningeal cells, oligodendrocytes, oligodendrocyte precursors, and ependymal astrocytes; results were substantially unchanged after removing WMH risk loci associated with BP (Supplementary Data 20).

To functionally characterize and prioritize individual WMH genomic risk loci we performed transcriptome-wide association studies (TWAS) using TWAS-Fusion[46], WMH-SNP association statistics from the main effects (EUR-only) and weights from 23

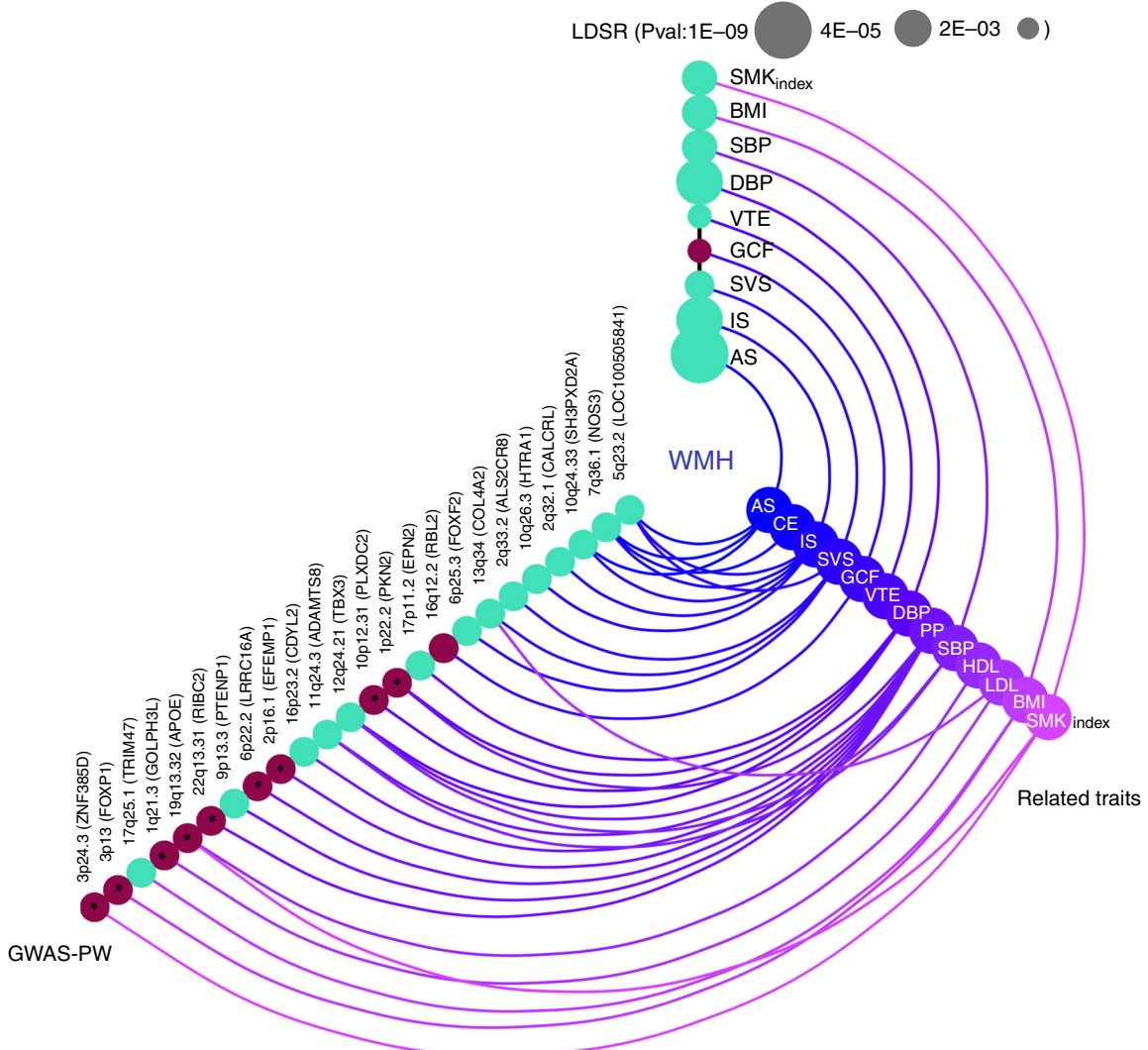

**Fig. 3 Shared genetic architecture of WMH at genome-wide and regional level Color coded for the direction of effect (Green: Positive genetic correlation; Red: Negative genetic correlation).** The LD-score regression (LDSR) axis shows evidence for genome-wide correlations (after Bonferroni correction for multiple testing $P < 3.6 \times 10^{-3}$, Methods), with the size of the nodes corresponding to the level of significance of the association. The GWAS-pairwise (PW) axis shows evidence for regional level overlap of association signals between WMH burden and related vascular and neurological traits (PPA3 ≥ 0.90, Methods). For any given region, the nearest gene (in brackets) to the top SNP associated with WMH is shown. Bivariate heritability estimator from summary statistics (ρ-HESS) was used to infer directionality of shared association signals (Methods) and asterisks denote an unexpected directionality of association. SBP systolic blood pressure, DBP diastolic blood pressure, PP pulse pressure, AS all stroke, IS ischemic stroke, SVS small vessel stroke, CE cardioembolic stroke, BMI body mass index, HDL high-density lipoprotein, LDL low-density lipoprotein, VTE venous thromboembolism, GCF general cognitive function, SMKindex lifetime smoking index, WMH white matter hyperintensity.

gene-expression reference panels from blood, arterial, and brain tissues (Supplementary Methods 2). We also included non-publicly available gene-expression weights from the dorsolateral prefrontal cortex (DLPFC) of 494 older community-dwelling participants (Methods)[47,48]. TWAS-Fusion identified 201 transcriptome-wide significant associations with WMH, conditionally significant on the predicted expression of a TWAS-associated gene, including 21 with splicing quantitative trait loci (sQTLs) regulating highly tissue-specific gene isoforms in DLPFC (Fig. 5, Supplementary Data 21). To rule out that observed associations reflect the random overlap between expression (eQTLs) and noncausal WMH risk variants, a colocalization analysis (COLOC)[49] was performed at each significant locus, to estimate the posterior probability of a shared causal variant (PP4 ≥ 75%) between the gene expression and trait association (Methods). Colocalization was observed for 96 TWAS significant eQTLs (48%, Fig. 5): of these, 54 mapped to 8 WMH genome-

wide risk loci and 16 expression-associated genes (eGenes), while 42 mapped to 12 distinct loci that were not genome-wide significant in the WMH GWAS and 23 eGenes. These additional putative WMH risk loci require confirmation in follow-up studies. Leveraging histone regulatory mark information from blood, arterial, and brain tissues (Methods, Supplementary Data 21), we observed that the majority (89%) of TWAS signals overlapped with enhancer and/or promoter elements, including eQTLs exhibiting weaker colocalization probability (PP4 < 75%). Larger WMH volume was associated with either upregulated or downregulated gene expression, the directionality being mostly consistent across broad tissue categories (Fig. 5). We found evidence for colocalization of WMH risk variants with eQTLs in brain tissues (28 eGenes), vascular tissues (15 eGenes), and blood (6 eGenes). Some eGenes (*KLHL24, NMT1, DCAKD, KANSL1, AMZ2P1*) showed evidence for colocalization in multiple tissues, and for some WMH risk loci (chr2q33.2, chr17q25.1,

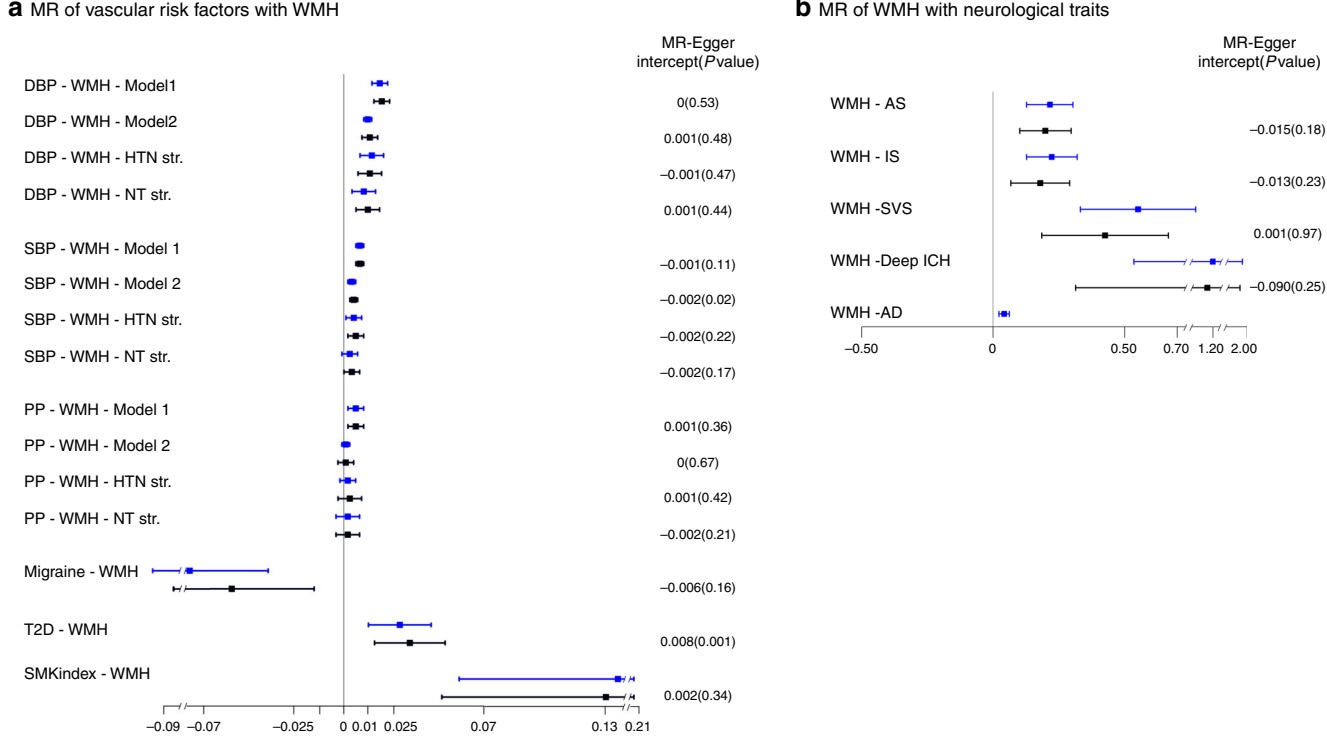

**Fig. 4 Mendelian randomization results of vascular risk factors with WMH burden (box A) and WMH burden with neurological traits (box B).** Point estimates and confidence intervals (blue) from the inverse-variance weighted (IVW) method are shown along with the point estimates and 95% confidence interval (black) from sensitivity analyses after filtering out potentially pleiotropic outlier variants. The intercept and p-value from the MR-Egger method is displayed on the far right (an intercept term significantly differing from zero at the conservative threshold of $P < 0.05$ suggests the presence of directional pleiotropy). SBP systolic blood pressure, DBP diastolic blood pressure, PP pulse pressure, HTN hypertensive, NT normotensive, Str. stratum, AS all stroke, IS ischemic stroke, SVS small vessel stroke, ICH intracerebral hemorrhage, AD Alzheimer's disease, T2D type II diabetes, SMKindex lifetime smoking index, WMH white matter hyperintensity, Model 1 Main effects adjusted for age, sex, principal components for population stratification, intracranial volume; Model 2 Model 1 + hypertension status.

chr17q21.31) colocalized variants associated with multiple eGenes with distinct tissue specificities (Fig. 5, Supplementary Data 21). WMH risk variants at the chr2q33 locus for instance showed evidence for colocalization with eQTLs for *NBEAL1* in nerve and arterial tissues, for *ICA1L* and *KRT8P15* in brain tissues, and for *CARF1* in right atrial appendage.

Among the 39 WMH eGenes with high colocalization probability (Fig. 5, Supplementary Data 21), 4 (*MAPT, CRHR1, CALCRL, NOTCH4*) are registered as targets of approved drugs in the DrugBank database and the Therapeutic Target Database (Supplementary Data 22).

## Discussion

This largest genetic study to date on complex SVD[13,14,16–18], leveraging genetic and brain-imaging information from 50,970 older community persons, triples the number of genetic loci associated with cerebral SVD and shows that this genetic risk results in detectable brain changes among asymptomatic young adults in their twenties. We further demonstrate the importance of higher BP as a risk factor for WMH even below clinical thresholds for HTN. MR analysis provides strong evidence for causal links of genetically determined WMH volume with risk of ischemic stroke, intracerebral haemorrhage, and Alzheimer-type dementia in later life. Importantly, we also provide insight into molecular pathways underlying SVD, highlighting relevant tissue and cell types, and suggest potential for genetic stratification of high-risk individuals and for genetically informed prioritization of drug targets for prevention trials.

Our approach focusing on the most common brain-imaging feature of SVD appears to be more powerful than GWAS of the small vessel stroke subtype to identify risk loci for SVD. Indeed, no new small vessel stroke risk locus was identified in MEGA-STROKE, the largest stroke GWAS meta-analysis to date[14]. We show a strong association between genetically determined WMH burden and risk of stroke in the general population, notably both risk of ischemic stroke and of intracerebral hemorrhage. While corroborating epidemiological observations[3,50], this has never been demonstrated using genetic instrumental variables, providing evidence for causality. This prompts greater caution with the common empirical prescription of antiplatelet therapy in persons with extensive WMH in the absence of clinical stroke[3], given the potential detrimental effects on intracerebral hemorrhage risk, and suggests the need for randomized clinical trials to determine the risk/benefit ratio of antiplatelet therapy in this setting.

The significant association we describe between genetically determined WMH burden and Alzheimer-type dementia also has potential important implications for prevention. It strengthens recent epidemiological evidence that WMH is associated not only with an increased risk of all and vascular dementia, but also of neurodegenerative Alzheimer-type dementia[3,51], providing for the first time evidence for causality using the WMH wGRS as an instrumental variable. Because of the proven ability to treat vascular risk factors, understanding and targeting the biological mechanisms of the vascular contribution to cognitive impairment and dementia, and specifically how cerebral SVD contributes to the molecular pathology of Alzheimer disease, are areas of intense research and clinical interest[52], especially given the current

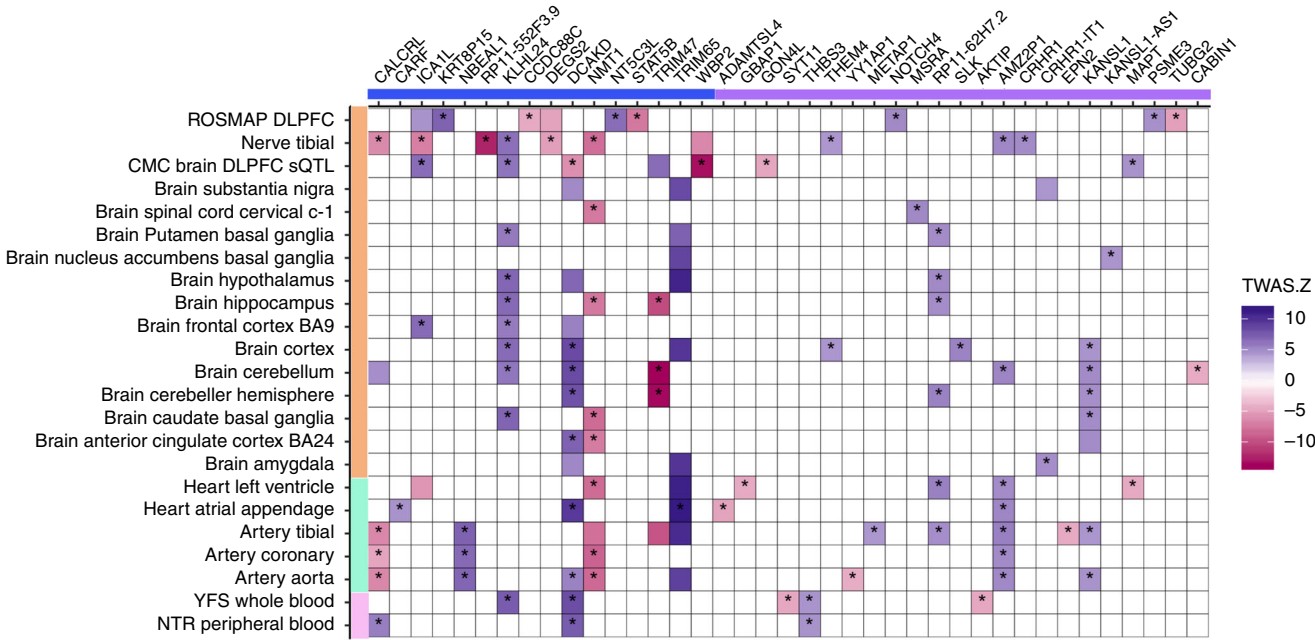

**Fig. 5 Transcriptome-wide association study of WMH and gene-expression datasets.** Only genes showing significant colocalization between the eQTL and the WMH risk variant in at least one tissue are shown. Susceptibility genes are depicted on the x-axis (blue: known; violet: novel), with tissue types of gene-expression datasets on the y-axis (orange: brain or peripheral nerve tissue; green: arterial/heart; pink: blood). Blue boxes correspond to WMH risk alleles being associated with upregulation (+) of gene expression in the corresponding tissues, while red boxes correspond to WMH risk alleles being associated with downregulation (−) of gene expression (color intensity corresponds to the magnitude of gene-expression effect size). Only significant TWAS associations at $P < 1.1 \times 10^{-5}$ are shown. Asterisks denote loci harboring a common causal variant associated with WMH and gene expression with high posterior probability using colocalization analyses (Methods; PP4 ≥ 0.75). ROSMAP religious order study and rush memory and aging project, DLPFC dorsolateral prefrontal cortex, CMC common mind consortium, BA Brodmann area, YFS young Finns study, NTR Netherlands twins register.

absence of other efficient therapies. Our results suggest that WMH should be considered a major target for preventative interventions, to mitigate not only the risk of stroke and vascular cognitive impairment but also of Alzheimer-type dementia, and support the rationale of innovative trials using WMH progression as a surrogate or intermediate endpoint for cognitive decline and dementia.

Over half of identified WMH risk loci are associated with higher BP levels. Moreover, using MR we provide evidence for a causal association between higher BP and larger WMH volume, notably even in participants without clinically defined HTN at the time of the MRI. Indeed, associations of genetically predicted SBP and DBP with WMH volume remained significant in participants without HTN, highlighting that higher levels of BP are likely causally associated with larger WMH volume even below BP thresholds typically used for the definition of HTN[43]. Considering the recent conclusions from the SPRINT-MIND trial suggesting that more drastic lowering of BP in persons with HTN is associated with slower progression of WMH volume and a lower risk of developing the combined outcome of mild cognitive impairment or dementia[53,54], our results suggest that trials to test a similar impact of intensive BP lowering in high-risk individuals who do not meet the current clinical thresholds for HTN could be warranted. We additionally show strong causal association between increased exposure to cigarette smoking over the lifetime (lifetime smoking index) and increased WMH burden, as has recently been described in relation with stroke risk[55], providing some additional evidence for the relevance of smoking cessation to prevent vascular brain injury and specifically SVD.

Importantly, a quarter of the identified WMH risk loci reflect molecular mechanisms that are not mediated by BP or other known vascular risk factors, two of these (NID2, VCAN), along with the COL4A2 and EFEMP1 locus, implicate genes encoding

matrisome proteins[56], involved in cell membrane structure and representing core components of the extracellular matrix (ECM). Converging evidence from experimental models for monogenic SVD suggest that perturbations of the matrisome play a central role in disease pathophysiology[57]. Our findings suggest that these could be relevant for sporadic forms of SVD. NID2 encodes nidogen, an ECM glycoprotein and a major component of basement membranes and is recognized as having a role in post-stroke angiogenesis[58,59]. Mutations in COL4A1/2, encoding collagen another basement membrane component already are known causing monogenic SVD[26]. VCAN, which we also found to be associated with white matter microstructure in young adults, encodes versican, a proteoglycan involved in cell adhesion and ECM assembly[60]. In CARASIL, a monogenic SVD caused by HTRA1 mutations, accumulation of versican in the thickened arterial wall was observed[27]. Versican also can form complexes that inhibit oligodendrocyte maturation and remyelination[61]. EFEMP1 encodes fibulin 3, an ECM glycoprotein localised in the basement membrane, and a proteolytic target of serine protease HTRA1[62]. Other previously unreported WMH risk loci that we have identified include KCNK2 that encodes Twik-related K+ channel (TREK1), a 2-pore-domain background ATP-sensitive potassium channel expressed throughout the central nervous system, more prominently in fetal than in adult brain. ECHDC3, near a distinct locus ($r^2 < 0.01$) previously implicated in Alzheimer disease[63]. MN1, which previously has been causally related to familial meningiomas[64], and XKR6, which has been associated with risk of systemic lupus erythematosus[65].

Our results provide important insight into the lifetime impact of genetic risk for SVD. Indeed, WMH risk variants observed in older adults were already associated with changes in DTI markers of white matter integrity in young adults in their early twenties. Of these, PSMD, a DTI metric recently described to be more

strongly correlated with cognitive performance in older persons (patients with sporadic or monogenic SVD and older community persons) than any other MRI-marker of SVD[30], was already showing nominal association with lower cognitive performance in young adults. This finding requires confirmation in future independent samples. The association of the WMH wGRS with subtle changes in white matter microstructure in young adults, if confirmed in independent samples, has potential important implications for the timing and paradigm of prediction and prevention of SVD progression and complications. It could reflect that biological pathways contributing to WMH at an older age already have a significant impact on brain microstructure in young adults, possibly reflecting a very early stage of SVD (typically characterized by reduced FA and increased MD and PSMD[33]). DTI changes and WMH have been suggested to be dependent physiological processes occurring within consecutive temporal windows in older patients with SVD[30,33,66]. Alternatively, observed associations might also reflect pleiotropy between SVD genes and genes influencing brain maturation, as the mean age of i-Share participants corresponds to the peak of white matter maturation[67]. On average FA tends to increase during childhood, adolescence, and early adulthood and then decline in middle-age, while the reverse is observed for MD[32,68]. Hence the association of the WMH wGRS with lower FA and higher MD could also reflect an impaired or delayed maturation or a premature aging process. The significant association of the WMH wGRS with RD but not $A_xD$ could potentially suggest that this is predominantly reflecting an impact on myelination of fiber tracts[69], in line with involvement of oligodendroglial dysfunction in early SVD pathology[70]. Future follow-up studies in a longitudinal setting are warranted to better understand the impact of genetically predicted WMH burden on the progression of white matter microstructural changes observed already in young adults and on their link with SVD and its complications.

Functional characterization revealed enrichment of WMH risk variants in regulatory marks in brain and neurosphere and in single-cell gene-expression levels in endothelial mural cells (as for clinical stroke)[71]. Gene prioritization using TWAS revealed that several WMH risk loci colocalized with eQTL for multiple genes with distinct tissue specificities. This pattern could potentially partly explain why association of such loci with WMH volume remained unchanged after controlling for the presence of HTN, although they were associated at genome-wide significant level with both BP and WMH. Of the 39 eGenes identified by TWAS four encode known drug targets. *MAPT* is a drug target under investigation for neurodegenerative disorders: the eQTL colocalizing with the WMH risk variant is an sQTL for the MAPT isoform in DLPFC and TWAS suggest that larger WMH volume is associated with upregulated *MAPT* expression. *CALCRL* encodes a component of the Calcitonin Gene Related Peptide receptor. TWAS suggest that lower abundance of the *CALCRL* transcript in arterial and nerve tissue and higher abundance in blood are associated with larger WMH volume. Monoclonal antibodies against *CALCRL* have recently been developed for the treatment of migraine[72].

We acknowledge limitations. We were underpowered for detecting additional risk variants for WMH after accounting for presence of HTN in the 2-df JMA gene-HTN interaction model. Recognizing that blood pressure is also highly variable and that a one-time blood pressure measurement may not reflect the long-term exposure of participants to high blood pressure levels, we conducted secondary analyses stratifying on quartiles of genetically predicted SBP and DBP levels, yielding similar results. In aggregate, a weighted genetic risk score of independent genome-wide significant WMH risk loci showed a significant 1-df interaction with HTN status in association with WMH volume,

suggesting that effect modification of genetic associations by HTN may exist, but that to discover them at the level of individual loci likely will require much larger samples. While we were able to use gene-expression data from many tissues for TWAS, such data are lacking for certain tissues that may be relevant for WMH (e.g., small brain vessels, microglia). Finally, our study population is predominantly of European ancestry (95%) limiting our ability to extrapolate our conclusions to other ancestry groups.

In conclusion we have identified 27 genetic risk loci for WMH volume, of which two thirds are not previously reported, and provided additional insight into their association with structural brain changes in very young adults, their clinical significance and the importance of high BP as a risk factor below clinical thresholds. Our results also point to molecular pathways underlying SVD that are not mediated by vascular risk factors and suggest potential for genetic stratification of high-risk individuals and genetically informed prioritization of drug targets for prevention trials.

## Methods

**Study population**. The study population comprised 23 population-based studies from the CHARGE consortium comprising a total of 24,182 individuals of European ($N = 21,666$) and African–American ($N = 2516$) ancestry, along with 26,788 community participants of European origin from the UK Biobank. In total, 50,970 participants were available for testing main genetic effects and 48,524 participants for the gene-hypertension interaction analysis (information on HTN status was missing in 2446 participants). Individuals with a history of stroke (or MRI-defined brain infarcts involving the cortical gray matter), or other pathologies that may influence the measurement of WMH (e.g., brain tumor, head trauma, etc.), at the time of MRI were excluded from analyses. Study participants in all participating cohorts gave written informed consent for phenotype quantification and use of genetic material (Supplementary Methods 1, Supplementary Table 1).

**Phenotypes**. MRI scans were obtained from scanners with field strengths ranging mostly from 1.5 to 3.0 Tesla and interpreted using a standardized protocol blinded to clinical or demographic features. In addition to T1 and T2 weighted scans along the axial plane, some cohorts included fluid-attenuated inversion recovery (FLAIR) and/or proton density (PD) sequences for better differentiation of WMH from cerebrospinal fluid. The vast majority of participating cohorts (>92% of all participants) used fully automated software to quantify WMH volume, with two cohorts using validated, visually guided semi-quantitative scales in older study subsets (Supplementary Table 1). WMH volume measures were inverse normal transformed to correct for skewness and account for differences in WMH quantification methods. Blood pressure measurements that are closest to the MRI scan were used to define HTN status. Participants with a SBP ≥ 140 mmHg, a DBP ≥ 90 mmHg, or taking antihypertensive medication were classified as having HTN.

**Genotyping and imputation**. Genome-wide genotyping platforms are described in Supplementary Data 2. Prior to imputation, sample-specific quality control (QC) on heterozygosity, missingness, gender mismatch, cryptic relatedness, and analysis of principal components (PC) for population stratification, as well as SNP-level QC on genotyping call rate and Hardy–Weinberg equilibrium were applied (Supplementary Data 2). Samples and SNPs passing the cohort-specific QC measures were then imputed to the 1000 genomes cosmopolitan panel phase 1 version 3 (1000 G p1v3) for CHARGE cohorts, while for the UK Biobank the dataset version 3 was imputed to the combined UK10K and Haplotype Reference Consortium (HRC) reference panels.

**Genome-wide association analyses**. Each participating study conducted ancestry-specific analyses using linear regression and assuming additive genetic effects under three models: (1) marginal genetic association test of WMH volume (SNP-main effect); (2) SNP-main effect adjusted for HTN status; (3) and joint association test of both SNP-main and SNP-by-HTN interaction effects in relation with WMH volume:

$$Y = \beta_0 + \beta_G \text{SNP} + \beta_C \text{Cov} + \varepsilon \tag{1}$$

$$Y = \beta_0 + \beta_G \text{SNP} + \beta_C \text{Cov} + \beta_{Env} \text{HTN} + \varepsilon \tag{2}$$

$$Y = \beta_0 + \beta_G \text{SNP} + \beta_C \text{Cov} + \beta_{GEnv} \text{SNP} \times \text{HTN} + \beta_{Env} \text{HTN} + \varepsilon \tag{3}$$

where SNP corresponds to the dosage of a given genetic (G) variant, Env is the dichotomous variable for HTN status, *Cov* is the vector of covariates, GEnv is the SNP-by-HTN interaction effect and $\beta$ values are the corresponding regression

coefficients and error covariance ($\varepsilon$) of $\beta$. The joint model (Model 3) provides effect estimates of G and GEnv, their robust standard errors (SE) and robust covariance matrices, and a joint $P$ value from a 2 degree-of-freedom (df) Wald test. Robust estimates of SEs and covariance matrices were used.

All analyses were adjusted for age, sex, PCs of population stratification and intracranial volume (ICV). Adjustment for ICV was not performed for studies using visual grading of WMH burden, as visual grades are inherently normalized for brain size[17] (Supplementary Data 1, Supplementary Methods 1).

**Genome-wide association meta-analyses.** A custom harmonization script along with the R package EasyQC[73] was used to perform the QC of cohort-specific GWAS results. SNPs with minor allele frequency (MAF) lower than 1%, poor imputation quality ($R^2 < 0.80$), or a product of MAF, $R^2$ and sample size less than 10, or 15 for the 2-df interaction analysis were excluded.

Inverse-variance weighted meta-analysis was conducted using METAL[74], first within each ancestry group followed by a meta-analysis of the ancestry-specific results. A patch implemented in the METAL[75] software was used to perform a 2-df joint meta-analysis (JMA) with inverse-variance weighting. For cohort-specific GWAS results with genomic inflation factors ($\lambda$) exceeding 1, genomic control (GC) correction was applied to correct for any residual population stratification. After meta-analysis only SNPs represented in more than half of participating studies and/or more than half of sample size and with no evidence of between-study heterogeneity ($PHet > 1 \times 10^{-4}$) were considered. Quantile-Quantile (QQ) plots of the P-values (observed versus expected) in the GWAS for the different models tested are presented along with the genomic inflation factor ($\lambda$) (Supplementary Fig. 2, Supplementary Data 3). Since heterogeneity in allelic effects that is specifically due to ancestry differences is not addressed by the traditional fixed-effects meta-analysis, a multiancestry meta-regression was carried out. For each variant, SNP-main allelic effects on WMH volume across GWAS were estimated in a linear regression framework weighting on the inverse of the variance of effect estimates and on the axes of genetic variation derived from pairwise allele frequency differences, as implemented in MR-MEGA[25]. It provides two heterogeneity estimates, one that is correlated with ancestry ($P_{Het}$-$ANC$) and accounted for in the meta-regression and the residual heterogeneity that is not due to population genetic differences ($P_{Het}$-$RES$).

In addition, association of the genome-wide significant WMH risk variants with WMH volume was tested after (i) stratification on hypertension status (all cohorts), and (ii) stratification on quartiles of SBP and DBP polygenic risk score distribution in the UK biobank, as described in the supplementary information (Supplementary Methods 2).

Across all association models, the power to reject the null hypothesis of no association at the genome-wide (GW) level was set at $P < 5 \times 10^{-8}$. Independent SNPs within genome-wide risk loci were determined by performing linkage disequilibrium (LD) based clumping implemented in PLINK using both a physical distance of ±1 megabase (Mb) and an LD threshold of $r^2 > 0.10$ from the index SNP of a given locus[76]. For constructing the LD matrix, ancestry-specific (European [EUR], African–American ancestry in South-West USA [ASW]) 1000 G p1v3 reference panels were used for ancestry-specific results and the merged (EUR +ASW) reference for multiancestry results. Stepwise conditional regression and joint analysis (cojo) implemented in GCTA[24] was performed to further validate the independent signals (based on the main-effects GWAS in Europeans only). GCTA-COJO additionally identifies signals with GW ($P < 5 \times 10^{-8}$) association level due to the LD adjusted joint effect of several neighbouring SNPs, selected based on an association priori of $P < 1 \times 10^{-7}$. Genotypes of 6489 unrelated European individuals from the Three City (3 C) study[77] were used to generate the LD matrix. Finally, gene-based association tests were conducted using MAGMA[28], with $P < 2.8 \times 10^{-6}$ as a gene-wide significance threshold. Gene regions with SNPs not reaching GW significance for WMH and/or not in LD ($r^2 < 0.10$) with the lead WMH SNP were considered as novel.

**WMH heritability estimates.** LD-score regression (LDSR) was used to distinguish polygenicity from confounding due to population stratification or cryptic relatedness[35] and to estimate the GW heritability by regressing the LD-score (measure of linked SNPs) against the chi-square association statistics of WMH volume from the European-only analysis. To address the infinitesimal-model assumption used by variance-component methods such as LDSR, we applied the heritability estimator from summary statistics (HESS)[29] to estimate local SNP-level heritability. HESS does not assume any effect size distribution and by weighted summation of the variant effect sizes and eigenvectors of the LD matrix provides variance explained by all SNPs at a given locus. Since the current GWAS sample size for the European-only analysis is smaller than the required size (>50,000) by HESS, GW heritability for WMH ($h^2 = 0.54 \pm 0.24$) from the 3C-Dijon study[9] was used to partition into each locus as suggested[29]. GWAS effects sizes were reinflated with the genomic inflation factor obtained from the GWAS summary statistics ($\lambda = 1.09$) to reduce potential downward bias of local SNP-level heritability and GW heritability estimates.

**Analysis of the lifetime impact of WMH risk variants.** We explored the association of the WMH wGRS (Supplementary Data 4) with MRI-markers of white

matter integrity in unrelated young adults participating in the i-Share cohort, the largest ongoing cohort study on student health (www.i-share.fr), using DTI markers. A WMH wGRS was constructed from 25 GW significant SNPs identified in European-only samples. High-quality MRI images and genome-wide genotypes were available in 1738 participants (Supplementary Methods 1, Supplementary Data 1). Briefly, white matter tracts were skeletonized with Tract-Based Spatial Statistics (TBSS) and a diffusion histogram analysis was performed, as described in the supplementary information (Supplementary Table 1), to derive DTI metrics measuring the integrity of the white matter microstructure, including fractional anisotropy (FA) and mean, radial and axial diffusivity (MD, RD, $A_xD$), as well as peak width of skeletonized mean diffusivity (PSMD). PSMD was calculated using a fully automated method via a shell script (www.psmd-marker.com) (Supplementary Table 1). A mixed linear model (MLM) was used to test the association of individual SNPs with each DTI trait, accounting for any sample substructure (admixture) and possible relatedness in the sample by using a genetic relationship matrix (GRM) as a random effect. The GRM was computed by implementing the MLMA-LOCO scheme in GCTA, where the SNP marker tested for association with a given outcome was excluded at each instance. MLMA-LOCO has been shown to better control false-positives over the standard mixed models especially in the presence of geographic population structure and cryptic relatedness[78]. The model was additionally adjusted for age, sex, ICV and the first four PCs of population stratification. The effect allele for each risk variant was defined as the allele associated with larger WMH volume. For associations with individual SNPs the significance threshold was set at $P < 2 \times 10^{-3}$ (0.05/25). The aggregate effect of 25 WMH risk variants with DTI metrics was estimated by using the GTX package in R[79].

The association of FA, MD, RD, $A_xD$, and PSMD with reaction time on the Stroop test, reflecting information processing speed, was examined using linear regression in i-Share participants who underwent both MRI and cognitive testing ($N = 1401$). Analyses were adjusted for age, sex, ICV, study-curriculum and ethnic origin. The association $p$ value was adjusted for the number of independent comparisons made ($n = 3$), estimated based on the correlation matrix between the DTI traits from i-Share and by applying the Matrix Spectral Decomposition (matSpDlite) method (http://neurogenetics.qimrberghofer.edu.au/matSpDlite/).

**Shared genetic architecture of WMH with related traits.** We systematically explored the genetic overlap of WMH SNP-main-effects (in the European-only analysis) with (i) neurological traits (any stroke, ischemic stroke, small vessel stroke, large artery stroke, cardioembolic stroke; any, deep, and lobar intracerebral hemorrhage; general cognitive function and Alzheimer-type dementia); and (ii) vascular risk factors and traits (SBP, DBP, PP, HDL-cholesterol, LDL-cholesterol, TG, BMI, T2D, HbA1c, SMKindex, VTE, and migraine). We acquired summary statistics of European-only analyses for these traits, using the latest largest GWAS, seeking collaboration with the relevant consortia when the data were not publicly available (Supplementary Table 2).

We first explored the association of lead WMH risk variants ($n = 27$) with related vascular and neurological traits. For each of the related traits, association statistics of SNPs falling in a window of ±250 kb around each of the lead WMH SNP were retrieved and SNPs satisfying the multiple testing threshold defined by correcting for the effective number of LD independent markers per locus, as implemented in Genetic Type 1 error calculator (GEC)[80], were retained. Only SNPs showing an association with a related vascular or neurological trait at P < $1.3 \times 10^{-4}$ (accounting for 14 independent traits and 27 independent loci) and in moderate to high LD with the lead WMH SNP ($r^2 > 0.50$) are reported. The correlation matrix estimated between the traits using individual-level data from 3 C study[77] was used to estimate the number of independent traits by applying the Matrix Spectral Decomposition (matSpDlite) method (http://neurogenetics.qimrberghofer.edu.au/matSpDlite/).

Using LDSR[81], genetic correlation estimates between WMH and the aforementioned neurological and vascular traits were obtained. A $p$ value < $3.6 \times 10^{-3}$ correcting for 14 independent phenotypes was considered significant. As genome-wide correlation estimates may miss significant correlations at the regional level (balancing-effect)[41], the Bayesian pairwise GWAS approach (GWAS-PW) was applied[36]. GWAS-PW identifies trait pairs with high posterior probability of association (PPA) with a shared genetic variant (model 3, PPA3 ≥ 0.90). To ensure that PPA3 is unbiased by sample overlap, fgwas v.0.3.6 was run on each pair of traits and the correlation estimated from regions with null association evidence (PPA < 0.20) was used as a correction factor[36]. Additionally, to estimate the directionality of associations with trait pairs in regions with PPA3 ≥ 0.90, HESS was used to estimate local genetic correlation[41].

**Mendelian randomization (MR).** For each vascular trait genetic variant (instrument) details were retrieved from the latest largest GWAS (Supplementary Table 2). Only independent SNPs ($r^2 < 0.10$, based on 1000 G EUR panel) reaching genome-wide significance were included as recommended[82]. Similarly, 25 independent WMH risk variants from the SNP-main effects were used as instruments to test the association of genetically predicted WMH volume with neurological traits. The putative causal effect ($\beta_{IVW}$) of an exposure on the outcome was estimated, using the inverse-variance weighting (IVW) method, by the weighted sum of the ratios of beta-coefficients from the SNP-outcome associations for each

variant ($j$) over corresponding beta-coefficients from the SNP-exposure associations ($\beta_j$). The ratio estimates from each genetic variant were averaged after weighting on the inverse variance ($W_j$) of $\beta_j$ across $L$ uncorrelated SNPs, implemented as an R package RadialMR (available through CRAN repositories)[40].

$$\beta IVW = \frac{\sum_{j=1}^{L} Wj\beta_j}{\sum_{j=1}^{L} Wj}$$

Effect alleles for each risk variant were defined as the allele associated with increase in the corresponding trait values. A $p$ value $< 3.6 \times 10^{-3}$ correcting for 14 independent traits was considered significant. Cochran's Q statistic was used to test for the presence of heterogeneity ($P_{Het} < 0.01$) due to horizontal pleiotropy that occurs when instruments exert an effect on the outcome and exposure through independent pathways[40]. Influential outlier SNPs that have the largest contribution to the global Cochran's Q statistic are identified by regressing the predicted causal estimate against the inverse-variance weights. After excluding the influential outlier SNPs, the IVW test was repeated along with MR-Egger regression[83]. Relative goodness of fit of the MR-Egger over the IVW approach was quantified using the $Q_R$ statistics, which is the ratio of the statistical heterogeneity around the MR-Egger fitted slope divided by the statistical heterogeneity around the IVW slope. A $Q_R$ close to 1 indicates that MR-Egger is not a better fit to the data and therefore offers no benefit over IVW[40]. Nonsignificant MR-Egger intercept was used as an indicator to formally rule out horizontal pleiotropy.

**Cell and tissue type enrichment analysis.** Association statistics from the WMH SNP-main effects (European-only) were used to test cell/tissue-specific enrichment. First we used the EPIGWAS software[44] and histone marks for promoters (H3K4me3) and enhancers (H3K4me1) from publicly available information on tissue-specific histone regulatory marks (Supplementary Methods 2). EPIGWAS calculates specificity scores for the lead WMH risk variant and its proxies ($r^2 \geq 0.80$, 1000 G EUR) based on the distance to the strongest ChIP-seq peak signal, and estimates enrichment significance by comparing the relative proximity and specificity of the test set with 10,000 sets of matched background (using permutation). Bonferroni correction for the number of histone marks tested was applied ($P < 2.5 \times 10^{-2}$). Second, we used MAGMA[28] (gene-property analysis) and differentially expressed gene sets from the single-cell transcriptomic (scRNA) data in mouse brain from the Karolinska Institute. MAGMA generates gene-level association statistics by combining SNP p-values in a specified window (10 kb upstream and 1.5 kb downstream of each gene) accounting for LD (1000 G EUR) and under a linear regression framework performs a one-sided test between the association of genes with WMH volume and cell specificity. Using the MAGMA.celltyping R package[45], scRNA expression values were obtained from five different mouse brain regions (neocortex, hippocampus, hypothalamus, striatum, and midbrain)[84]. A gene-expression specificity metric for each cell-type was calculated by dividing the expression level in a given cell type by the sum of the expression levels from all cell types (i.e., genes with high expression levels in two or more cell types will get a lower specificity measure than genes with high expression levels in a single-cell type), followed by binning the metric value to 40 equally sized bins. The MAGMA one-sided test was then used to test for enrichment between the top 10 percentile bins (bins with higher cell-specificity) in each cell type. Bonferroni correction for the number of cell types tested was applied ($P < 2.1 \times 10^{-3}$).

**Transcriptome-wide association study and colocalization.** We performed transcriptome-wide association studies (TWAS) using the association statistics from the WMH SNP-main effects (European-only) and weights from 22 publicly available gene-expression reference panels (Supplementary Methods 2) from blood (Netherlands Twin Registry, NTR; Young Finns Study, YFS), arterial (Genotype-Tissue Expression, GTEx), brain (GTEx, CommonMind Consortium, CMC) and peripheral nerve tissues (GTEx). For each gene in the reference panel, precomputed SNP-expression weights in the 1-Mb window were obtained (Supplementary Methods 2), including the highly tissue-specific splicing QTL (sQTL) information on gene isoforms in the dorsolateral prefrontal cortex (DLPFC) derived from the CMC. Additionally, non-publicly available gene-expression weights from the DLPFC of 494 older individuals from two large community-based studies (the Religious Order Study [ROS][48] and the Rush Memory Aging Project [MAP][47] were obtained. TWAS-Fusion[46] was used to estimate the TWAS Z score (association statistic between predicted expression and WMH), derived from the SNP-expression weights, SNP-WMH effect estimates and the SNP correlation matrix. Transcriptome-wide (TW) significant genes (eGenes) and the corresponding QTLs (eQTLs) were determined using Bonferroni correction in each reference panel, based on the average number of features (4360 genes) tested across all the reference panels[46]. eGene regions with eQTLs not reaching genome-wide significance in association with WMH, and not in LD ($r^2 < 0.01$) with the lead SNP for genome-wide significant WMH risk loci, were considered as novel. Finally, a colocalization analysis (COLOC)[49] was carried out at each locus to estimate the posterior probability of a shared causal variant (PP4 ≥ 0.75) between the gene expression and trait association, using a prior probability of $1.1 \times 10^{-5}$ for the WMH association. Furthermore, functional validation of the eGenes was performed by testing for positional overlap of the best eQTLs from TWAS with enhancer (H3K4me1) and/or promoter (H3K4me3) elements across a broad category of relevant tissue types

(blood, brain/neurological, heart/arterial) using Haploreg V4.1[85]. A value of 1 was assigned to eQTLs with regulatory epigenome overlap in at least one tissue.

**Drug-target enrichment.** The Genome for REPositioning drugs (GREP) tool[86] was used to quantify the enrichment of eGenes emerging from the TWAS with high probability of colocalization (PP4 ≥ 0.75) in the curated drug-target list classified based on the International Classification of Diseases 10 (ICD10). GREP provides as an output the names of the drug(s) targeting a given gene set along with the disease category. Moreover, by performing a series of Fisher's exact tests GREP formally tests whether the gene set is enriched in genes targeted by drugs in a specific clinical indication category to treat a certain disease or condition.

**Reporting summary.** Further information on research design is available in the Nature Research Life Sciences Reporting Summary linked to this article.

## Data availability

Summary statistics for the GWAS meta-analysis of the CHARGE cohorts and the UK Biobank on WMH burden main-effects generated and analyzed in the downstream analyses are deposited in a public repository (dbGAP: https://www.ncbi.nlm.nih.gov/gap/) under the accession number: phs002227.v1.p1. All other data supporting the findings of this study are available either within the article, the supplementary information and supplementary data files, or from the authors upon reasonable request. Publicly available resources used for this study were the UK Biobank [http://www.ukbiobank.ac.uk/]; Gene-expression weights for TWAS [http://gusevlab.org/projects/fusion/]; Magma.Celltyping [https://github.com/NathanSkene/MAGMA_Celltyping]; Histone regulatory marks [http://egg2.wustl.edu/roadmap/data/byFileType/peaks/consolidated/narrowPeak/]. Genome-wide summary statistics for other complex disorders were downloaded from public repositories [Supplementary Methods 2 for URL links].

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

## Acknowledgements

We thank all the participating cohorts from the CHARGE consortium and the UK biobank participants for contributing to this study. Detailed study-specific acknowledgements are provided in the supplementary note.

## Author contributions

W.T.L., L.J.L., M.L., S.Seshadri, C.T., H.H.A., P.M.M., M.F., and S.D. jointly supervised research. M.S., H.Suzuki, XJ, C.S., and T.E.E. contributed equally. M.S., W.T.L., L.J.L., M.L., S.Seshadri, C.T., H.H.A., P.M.M., M.F., and S.D. Designed and conceived the study. M.S., H.Suzuki, XJ, C.S., T.E.E., J.C.B., G.E., S.Sakaue, N.T., M.H., W.Z., N.J.A., E.H., L.R.Y., S.P.H., R.B.K., E.B.V., R.E.M., S.T., A.M., Y.S., C.L.S., G.B., L.P., A.Tsuchida, L.Z., S.Schilling, S.Sigurdsson, R.F.G., C.E.L., N.T.A., O.L.L., J.A.S., M.C.V., J.v., M.J.W., M.J.K., M.D., R.J.T., C.B., M.G.D., A.V.S., D.S.K., P.J.S., D.A.E., J.I.R., A.S.B., S.M.M., M.B., J.T., D.J.S., M.W.V., K.W., W.J.N., A.S., E.B., S.Sidney, S.T.T., G.D., A.Thalamuthu, U.V., M.A.V., N.R.B., J.D., M.E.B., D.A., A.Teumer, P.A., J.B.K., R.B., I.J.D., P.R.S., H.B., J.J., Y.T., K.S., S.M., K.Y., M.N., Y.K., F.M., B.M.P., D.A.B., P.L.D., T.H.M., P.S.S., R.S., H.R.W., E.E., D.T., M.A.I., W.W., C.D., V.K.S., J.Wouter.J., E.P.S., S.I.R.K., Y.O., B.M., J.M.W., P.A.N., K.A.M., H.J.G., H.Schmidt, and V.G. conducted cohort-wise GWAS analysis. M.S. performed shared genetic architecture and functional characterization analysis. M.S., C.B., and A.S. performed lifetime impact analysis. M.S. and S.Sakaue performed drug-target enrichment. M.S., W.T.L., L.J.L., M.L., S.Seshadri, C.T., H.H.A., P.M.M., M.F., and S.D. wrote and edited the manuscript.

## Competing interests

The authors declare no competing interests.

## Additional information

Muralidharan Sargurupremraj[1,216], Hideaki Suzuki[2,3,4,216], Xueqiu Jian[5,6,216], Chloé Sarnowski[7,216], Tavia E. Evans[8,9,216], Joshua C. Bis[10,216], Gudny Eiriksdottir[11], Saori Sakaue[12,13,14], Natalie Terzikhan[15], Mohamad Habes[6,16,17], Wei Zhao[18], Nicola J. Armstrong[19], Edith Hofer[20,21], Lisa R. Yanek[22], Saskia P. Hagenaars[23,24], Rajan B. Kumar[25], Erik B. van den Akker[26,27,28], Rebekah E. McWhirter[29,30], Stella Trompet[31,32], Aniket Mishra[1], Yasaman Saba[1,33], Claudia L. Satizabal[6,34,35], Gregory Beaudet[36], Laurent Petit[36], Ami Tsuchida[36], Laure Zago[36], Sabrina Schilling[1], Sigurdur Sigurdsson[11], Rebecca F. Gottesman[37], Cora E. Lewis[38], Neelum T. Aggarwal[39], Oscar L. Lopez[40], Jennifer A. Smith[18,41], Maria C. Valdés Hernández[23,42,43], Jeroen van der Grond[44], Margaret J. Wright[45,46], Maria J. Knol[15], Marcus Dörr[47,48], Russell J. Thomson[30,49], Constance Bordes[1], Quentin Le Grand[1], Marie-Gabrielle Duperron[1], Albert V. Smith[11], David S. Knopman[50], Pamela J. Schreiner[51], Denis A. Evans[52], Jerome I. Rotter[53], Alexa S. Beiser[7,34,35], Susana Muñoz Maniega[23,42], Marian Beekman[26], Julian Trollor[54,55], David J. Stott[56], Meike W. Vernooij[9,15], Katharina Wittfeld[57], Wiro J. Niessen[9,58],

Aicha Soumaré[1], Eric Boerwinkle[59], Stephen Sidney[60], Stephen T. Turner[61], Gail Davies[21,62], Anbupalam Thalamuthu[53], Uwe Völker[63], Mark A. van Buchem[43], R. Nick Bryan[64], Josée Dupuis[6,32], Mark E. Bastin[21,41], David Ames[65,66], Alexander Teumer[15,47], Philippe Amouyel[67,68], John B. Kwok[69,70], Robin Bülow[71], Ian J. Deary[21,62], Peter R. Schofield[70,72], Henry Brodaty[53,73], Jiyang Jiang[53], Yasuharu Tabara[74], Kazuya Setoh[75], Susumu Miyamoto[75], Kazumichi Yoshida[75], Manabu Nagata[75], Yoichiro Kamatani[76], Fumihiko Matsuda[74], Bruce M. Psaty[77,78], David A. Bennett[79], Philip L. De Jager[80,81], Thomas H. Mosley[82], Perminder S. Sachdev[53,83], Reinhold Schmidt[18], Helen R. Warren[84,85], Evangelos Evangelou[86,87], David-Alexandre Trégouët[1], International Network against Thrombosis (INVENT) Consortium*, International Headache Genomics Consortium (IHGC)*, Mohammad A. Ikram[15], Wei Wen[54], Charles DeCarli[88], Velandai K. Srikanth[30,89], J. Wouter Jukema[32], Eline P. Slagboom[26], Sharon L. R. Kardia[18], Yukinori Okada[12,13,90], Bernard Mazoyer[36], Joanna M. Wardlaw[23,42,43,91], Paul A. Nyquist[92,93], Karen A. Mather[54,72], Hans J. Grabe[94,95], Helena Schmidt[33], Cornelia M. Van Duijn[96], Vilmundur Gudnason[11,97], William T. Longstreth Jr[98], Lenore J. Launer[99,100], Mark Lathrop[101,217], Sudha Seshadri[6,34,35,217], Christophe Tzourio[1,102,217], Hieab H. Adams[8,9,217], Paul M. Matthews[4,103,104,217], Myriam Fornage[5,217✉] & Stéphanie Debette[1,35,105,217✉]

[1]University of Bordeaux, Inserm, Bordeaux Population Health Research Center, team VINTAGE, UMR 1219, 33000 Bordeaux, France. [2]Tohoku Medical Megabank Organization, Tohoku University, 2-1, Seiryo, Aoba, Sendai 980-8573, Japan. [3]Department of Cardiovascular Medicine, Tohoku University Hospital, 1-1, Seiryo, Aoba, Sendai 980-8574, Japan. [4]Department of Brain Sciences, Imperial College London, London W12 0NN, UK. [5]University of Texas Health Science Center at Houston McGovern Medical School, Houston, TX 77030, USA. [6]Glenn Biggs Institute for Alzheimer's & Neurodegenerative Diseases, University of Texas Health Sciences Center, San Antonio, TX 78229, USA. [7]Department of Biostatistics, Boston University School of Public Health, Boston, MA 02118, USA. [8]Department of Clinical Genetics, Erasmus MC, 3015 GE Rotterdam, The Netherlands. [9]Department of Radiology & Nuclear Medicine, Erasmus MC, 3015 GE Rotterdam, The Netherlands. [10]Cardiovascular Health Research Unit, Department of Medicine, University of Washington, Seattle, WA 98101, USA. [11]Icelandic Heart Association, IS-201 Kópavogur, Iceland. [12]Department of Statistical Genetics, Osaka University Graduate School of Medicine, Suita 565-0871, Japan. [13]Laboratory for Statistical Analysis, RIKEN Center for Integrative Medical Sciences, Tsurumi-ku, Yokohama City, Kanagawa 230-0045, Japan. [14]Department of Allergy and Rheumatology, Graduate School of Medicine, the University of Tokyo, Tokyo 113-0033, Japan. [15]Department of Epidemiology, Erasmus MC, 3015 GE Rotterdam, The Netherlands. [16]Department of Radiology, Perelman School of Medicine, University of Pennsylvania, Philadelphia, PA 19104, USA. [17]Institute for Community Medicine, University Medicine Greifswald, 17475 Greifswald, Germany. [18]Department of Epidemiology, School of Public Health, University of Michigan, Ann Arbor, MI 48109-2029, USA. [19]Mathematics and Statistics, Murdoch University, Murdoch, WA 6150, Australia[20]Clinical Division of Neurogeriatrics, Department of Neurology, Medical University of Graz, 8036 Graz, Austria. [21]Institute for Medical Informatics, Statistics and Documentation, Medical University of Graz, 8036 Graz, Austria. [22]GeneSTAR Research Program, Division of General Internal Medicine, Department of Medicine, Johns Hopkins University School of Medicine, Baltimore, MD 21205, USA. [23]Centre for Cognitive Ageing and Cognitive Epidemiology, University of Edinburgh, Edinburgh EH8 9JZ, UK. [24]Social Genetic and Developmental Psychiatry Research Centre, Institute of Psychiatry, Psychology & Neuroscience, King's College London, London SE5 8AF, UK. [25]Department of Public Health Sciences, University of California at Davis, Davis, CA 95616, USA. [26]Section of Molecular Epidemiology, Biomedical Sciences, Leiden university Medical Center, 2333 ZA Leiden, The Netherlands. [27]Pattern Recognition & Bioinformatics, Delft University of Technology, Delft, NL 2629 HS, USA. [28]Leiden Computational Biology Centre, Leiden University Medical Centre, 2333 ZA Leiden, The Netherlands. [29]Centre for Law and Genetics, Faculty of Law, University of Tasmania, Hobart, TAS 7005, Australia. [30]Menzies Institute for Medical Research, University of Tasmania, Hobart, TAS 7000, Australia. [31]Department of Internal Medicine, section of gerontology and geriatrics, Leiden University Medical Center, 2333 ZA Leiden, The Netherlands. [32]Department of Cardiology, Leiden University Medical Center, 2333 ZA Leiden, The Netherlands. [33]Gottfried Schatz Research Center, Department of Molecular Biology and Biochemistry, Medical University of Graz, 8010 Graz, Austria. [34]Boston University and the NHLBI's Framingham Heart Study, Boston, MA 02215, USA. [35]Department of Neurology, Boston University School of Medicine, Boston, MA 02118, USA. [36]University of Bordeaux, IMN, UMR 5293, 33000 Bordeaux, France. [37]Johns Hopkins University School of Medicine, Baltimore, MD 21205, USA. [38]University of Alabama at Birmingham School of Medicine, Birmingham, AL 35233, USA. [39]Department of Neurological Sciences, Rush University Medical Center, Chicago, IL 60612, USA. [40]Departments of Neurology and Psychiatry, University of Pittsburgh, Pittsburgh, PA 15213, USA. [41]Survey Research Center, Institute for Social Research, University of Michigan, Ann Arbor, MI 48104, USA. [42]Centre for Clinical Brain Sciences, University of Edinburgh, Edinburgh EH16 4SB, UK. [43]Row Fogo Centre for Ageing and The Brain, University of Edinburgh, Edinburgh EH8 9JZ, UK. [44]Department of Radiology, Leiden University medical Center, 2333 ZA Leiden, The Netherlands. [45]Queensland Brain Institute, The University of Queensland, St Lucia, QLD 4072, Australia. [46]Centre for Advanced Imaging, The University of Queensland, St Lucia, QLD 4072, Australia. [47]Department of Internal Medicine B, University Medicine Greifswald, 17475 Greifswald, Germany. [48]DZHK (German Center for Cardiovascular Research), partner site Greifswald, 17475 Greifswald, Germany. [49]Centre for Research in Mathematics and Data Science, Western Sydney University, Penrith, NSW 2751, Australia. [50]Mayo Clinic, Rochester, MN 55905, USA. [51]University of Minnesota School of Public Health, Minneapolis, MN 55455, USA. [52]Department of Internal Medicine, Rush University Medical Center, Chicago, IL 60612, USA. [53]Institute for Translational Genomics and Population Sciences, Los Angeles Biomedical Research Institute and Pediatrics at Harbor-UCLA Medical Center, Torrance, CA 90502, USA. [54]Centre for Healthy Brain Ageing, School of Psychiatry, University of New South Wales, Sydney, NSW 2052, Australia. [55]Department of Developmental Disability Neuropsychiatry, School of Psychiatry, University of New South Wales, Sydney, NSW 2052, Australia. [56]Institute of Cardiovascular and Medical Sciences, College of Medical, Veterinary and Life Sciences, University of Glasgow, Glasgow G12 8QQ, UK. [57]German Center for Neurodegenerative Diseases (DZNE), Rostock/Greifswald, 17489 Greifswald, Germany. [58]Faculty of Applied Sciences, Delft University of Technology, Delft, NL 2629 HS, USA. [59]University of Texas Health Science Center at Houston School of Public Health,

Houston, TX 77030, USA. [60]Kaiser Permanente Division of Research, Oakland, CA 94612, USA. [61]Division of Nephrology and Hypertension, Mayo Clinic, Rochester, MN 55905, USA. [62]Department of Psychology, University of Edinburgh, Edinburgh EH8 9JZ, UK. [63]Interfaculty Institute for Genetics and Functional Genomics, University Medicine Greifswald, 17475 Greifswald, Germany. [64]The University of Texas at Austin Dell Medical School, Austin, TX 78712, USA. [65]National Ageing Research Institute Royal Melbourne Hospital, Parkville, VIC 3052, Australia. [66]Academic Unit for Psychiatry of Old Age, University of Melbourne, St George's Hospital, Kew, VIC 3101, Australia. [67]Inserm U1167, 59000 Lille, France. [68]Department of Epidemiology and Public Health, Pasteur Institute of Lille, 59000 Lille, France. [69]Brain and Mind Centre - The University of Sydney, Camperdown, NSW 2050, Australia. [70]School of Medical Sciences, University of New South Wales, Sydney, NSW 2052, Australia. [71]Department of Diagnostic Radiology and Neuroradiology, University Medicine Greifswald, 17489 Greifswald, Germany. [72]Neuroscience Research Australia, Randwick, NSW 2031, Australia. [73]Dementia Centre for Research Collaboration, University of New South Wales, Sydney, NSW 2052, Australia. [74]Center for Genomic Medicine, Kyoto University Graduate School of Medicine, Kyoto 606-8501, Japan. [75]Department of Neurosurgery, Kyoto University Graduate School of Medicine, Kyoto 606-8501, Japan. [76]Laboratory for Statistical and Translational Genetics, RIKEN Center for Integrative Medical Sciences, Tsurumi-ku, Yokohama City, Kanagawa 230-0045, Japan. [77]Departments of Epidemiology, Medicine and Health Services, University of Washington, Seattle, WA 98195, USA. [78]Kaiser Permanente Washington Health Research Institute, Seattle, WA 98101, USA. [79]Rush Alzheimer's Disease Center, Rush University Medical Center, Chicago, IL 60612, USA. [80]Center for Translational and Computational Neuroimmunology, Department of Neurology, Columbia University Medical Center, New York, NY 10032, USA. [81]Program in Population and Medical Genetics, Broad Institute of MIT and Harvard, Cambridge, MA 02142, USA. [82]Memory Impairment and Neurodegenerative Dementia (MIND) Center, University of Mississippi Medical Center, Jackson, MS 39216, USA. [83]Neuropsychiatric Institute, Prince of Wales Hospital, Sydney, NSW 2031, Australia. [84]William Harvey Research Institute, Barts and The London School of Medicine and Dentistry, Queen Mary University of London, London E1 4NS, UK. [85]National Institute for Health Research Barts Cardiovascular Biomedical Research Unit, Queen Mary University of London, London EC1M 6BQ, UK. [86]Department of Epidemiology and Biostatistics, School of Public Health, Imperial College London, London SW7 2AZ, UK. [87]Department of Hygiene and Epidemiology, University of Ioannina Medical School, Ioannina, Mpizani 455 00, Greece. [88]Department of Neurology and Center for Neuroscience, University of California at Davis, Sacramento, CA 95817, USA. [89]Peninsula Clinical School, Central Clinical School, Monash University, Melbourne, VIC 3004, Australia. [90]Laboratory of Statistical Immunology, Immunology Frontier Research Center (WPI-IFReC), Osaka University, Suita 565-0871 Osaka, Japan. [91]MRC UK Dementia Research Institute at the University of Edinburgh, Edinburgh EH8 9YL, UK. [92]Department of Neurology, Johns Hopkins School of Medicine, Baltimore, MD 21205, USA. [93]General Internal Medicine, Johns Hopkins School of Medicine, Baltimore, MD 21205, USA. [94]Department of Psychiatry and Psychotherapy, University Medicine Greifswald, 17475 Greifswald, Germany. [95]German Center for Neurodegenerative Diseases (DZNE), Rostock/Greifswald, 17475 Greifswald, Germany. [96]Nuffield Department of Population Health, University of Oxford, Oxford OX3 7LF, UK. [97]University of Iceland, Faculty of Medicine, 101 Reykjavík, Iceland. [98]Departments of Neurology and Epidemiology, University of Washington, Seattle, WA 98104-2420, USA. [99]Laboratory of Epidemiology, Demography, and Biometry, National Institute of Aging, The National Institutes of Health, Bethesda, MD 20892, USA. [100]Intramural Research Program/National Institute on Aging/National Institutes of Health, Bethesda, MD 20892, USA. [101]University of McGill Genome Center, Montreal, QC H3A 0G1, Canada. [102]CHU de Bordeaux, Pole de santé publique, Service d'information médicale, 33000 Bordeaux, France. [103]UK Dementia Research Institute, London WC1E 6BT, UK. [104]Data Science Institute, Imperial College London, London SW7 2AZ, UK. [105]Department of Neurology, CHU de Bordeaux, 33000 Bordeaux, France. [216]These authors contributed equally: Muralidharan Sargurupremraj, Hideaki Suzuki, Xueqiu Jian, Chloé Sarnowski, Tavia E. Evans, Joshua C. Bis. [217]These authors jointly supervised this work: Mark Lathrop, Sudha Seshadri, Christophe Tzourio, Hieab H. Adams, Paul M. Matthews, Myriam Fornage, Stéphanie Debette. *Lists of authors and their affiliations appear at the end of the paper. ✉email: myriam.fornage@uth.tmc.edu; stephanie.debette@u-bordeaux.fr

## International Network against Thrombosis (INVENT) Consortium

Philippe Amouyel[106,107], Mariza de Andrade[108], Saonli Basu[109], Claudine Berr[110], Jennifer A. Brody[111], Daniel I. Chasman[112], Jean-Francois Dartigues[113], Aaron R. Folsom[114], Marine Germain[115], Hugoline de Haan[116], John Heit[117], Jeanine Houwing-Duitermaat[118], Christopher Kabrhel[119], Peter Kraft[120], Grégoire Legal[121,122], Sara Lindström[120], Ramin Monajemi[118], Pierre-Emmanuel Morange[123,124,125], Bruce M. Psaty[111,126], Pieter H. Reitsma[127], Paul M. Ridker[128], Lynda M. Rose[129], Frits R. Rosendaal[116], Noémie Saut[123,124,125], Eline Slagboom[130], David Smadja[131,132,133], Nicholas L. Smith[126,134,135], Pierre Suchon[123,124,125], Weihong Tang[114], Kent D. Taylor[136], David-Alexandre Trégouët[115], Christophe Tzourio[115], Marieke C. H. de Visser[127], Astrid van Hylckama Vlieg[116], Lu-Chen Weng[114] & Kerri L. Wiggins[134]

[106]Institut Pasteur de Lille, Université de Lille Nord de France, INSERM UMR_S 744, Lille, France. [107]Centre Hospitalier Régional Universitaire de Lille, Lille, France. [108]Division of Biomedical Statistics and Informatics Mayo Clinic, Rochester, MN, USA. [109]University of Minnesota, Division of Biostatistics, Minneapolis, MN, USA. [110]INSERM Research Unit U1061, University of Montpellier I, Montpellier, France. [111]Cardiovascular Health Research Unit, Departments of Medicine, Epidemiology, and Health Services, University of Washington, Seattle, WA, USA. [112]Division of Preventive Medicine, Brigham and Women's Hospital and Harvard Medical School, Boston, MA 02215, USA. [113]INSERM Research Center U897, University of Bordeaux, Bordeaux, France. [114]University of Minnesota, Division of Epidemiology and Community Health Minneapolis, Minneapolis, MN, USA. [115]Institut National pour la Santé et la Recherche Médicale (INSERM), Unité Mixte de Recherche en Santé (UMR_S) 1219, Bordeaux Population Health Research Center, 33076 Bordeaux, France. [116]Department of Thrombosis and Hemostasis, Leiden University Medical Center, Leiden, The Netherlands; Department of Clinical Epidemiology, Leiden University Medical Center, Leiden, The Netherlands. [117]Division of Cardiovascular Diseases, Mayo Clinic, Rochester, MN, USA. [118]Department of Medical Statistics and Bioinformatics, Leiden University Medical Center, 2300 RC Leiden, The Netherlands. [119]Department of Emergency Medicine, Massachusetts General Hospital, Channing Network Medicine, Harvard Medical School, Boston, MA 2114, USA. [120]Program in Genetic Epidemiology and Statistical Genetics, Department of Epidemiology, Harvard School of Public Health, Boston, MA 2115, USA. [121]Université de Brest, EA3878 and CIC1412, Brest, France. [122]Ottawa Hospital Research Institute at the University of Ottawa, Ottawa, ON, Canada. [123]Laboratory of Haematology, La Timone Hospital, F-13385 Marseille, France. [124]INSERM, UMR_S

1062, Nutrition Obesity and Risk of Thrombosis, F-13385 Marseille, France. [125]Aix-Marseille University, UMR_S 1062, Nutrition Obesity and Risk of Thrombosis, F-13385 Marseille, France. [126]Group Health Research Institute, Group Health Cooperative, Seattle, WA 98101, USA. [127]Einthoven Laboratory for Experimental Vascular Medicine, Department of Thrombosis and Hemostasis, Leiden University Medical Center, 2300 RC Leiden, The Netherlands. [128]Division of Preventive Medicine, Brigham and Women's Hospital and Harvard Medical School, Boston, MA 02215, USA. [129]Division of Preventive Medicine, Brigham and Women's Hospital, Boston, MA 02215, USA. [130]Department of Molecular Epidemiology, Leiden University Medical Center, 2300 RC Leiden, The Netherlands. [131]Université Paris Descartes, Sorbonne Paris Cité, Paris, France. [132]AP-HP, Hopital Européen Georges Pompidou, Service 'hématologie Biologique, Paris, France. [133]INSERM, UMR_S 1140, Faculté de Pharmacie, Paris, France. [134]Department of Epidemiology, University of Washington, Seattle, WA 98195, USA. [135]Seattle Epidemiologic Research and Information Center, VA Office of Research and Development, Seattle, WA 98108, USA. [136]Los Angeles Biomedical Research Institute and Department of Pediatrics, Harbor-UCLA Medical Center, Torrence, CA 90502, USA.

## International Headache Genomics Consortium (IHGC)

Padhraig Gormley[137,138,139,140], Verneri Anttila[138,139,141], Bendik S. Winsvold[142,143,144], Priit Palta[145], Tonu Esko[138,146,147], Tune H. Pers[138,147,148,149], Kai-How Farh[138,141,150], Ester Cuenca-Leon[137,138,139,151], Mikko Muona[145,152,153,154], Nicholas A. Furlotte[155], Tobias Kurth[156,157], Andres Ingason[158], George McMahon[159], Lannie Ligthart[160], Gisela M. Terwindt[161], Mikko Kallela[162], Tobias M. Freilinger[163,164], Caroline Ran[165], Scott G. Gordon[166], Anine H. Stam[161], Stacy Steinberg[158], Guntram Borck[167], Markku Koiranen[168,169], Lydia Quaye[170], Hieab H. H. Adams[171,172], Terho Lehtimäki[173], Antti-Pekka Sarin[145], Juho Wedenoja[174], David A. Hinds[155], Julie E. Buring[157,175], Markus Schürks[176], Paul M. Ridker[157,175], Maria Gudlaug Hrafnsdottir[177], Hreinn Stefansson[158], Susan M. Ring[159], Jouke-Jan Hottenga[160], Brenda W. J. H. Penninx[178], Markus Färkkilä[162], Ville Artto[162], Mari Kaunisto[145], Salli Vepsäläinen[162], Rainer Malik[163], Andrew C. Heath[179], Pamela A. F. Madden[179], Nicholas G. Martin[166], Grant W. Montgomery[166], Mitja Kurki[137,138,139], Mart Kals[146], Reedik Mägi[146], Kalle Pärn[146], Eija Hämäläinen[145], Hailiang Huang[138,139,141], Andrea E. Byrnes[138,139,141], Lude Franke[180], Jie Huang[140], Evie Stergiakouli[159], Phil H. Lee[137,138,139], Cynthia Sandor[181], Caleb Webber[181], Zameel Cader[182,183], Bertram Muller-Myhsok[184], Stefan Schreiber[185], Thomas Meitinger[186], Johan G. Eriksson[187,188], Veikko Salomaa[188], Kauko Heikkilä[189], Elizabeth Loehrer[171,190], Andre G. Uitterlinden[191], Albert Hofman[171], Cornelia M. van Duijn[171], Lynn Cherkas[170], Linda M. Pedersen[142], Audun Stubhaug[192,193], Christopher S. Nielsen[192,194], Minna Männikkö[168,169], Evelin Mihailov[146], Lili Milani[146], Hartmut Göbel[195], Ann-Louise Esserlind[196], Anne Francke Christensen[196], Thomas Folkmann Hansen[197], Thomas Werge[198,199,200], Jaakko Kaprio[145,174,201], Arpo J. Aromaa[188], Olli Raitakari[202,203], M. Arfan Ikram[171,172,203,204], Tim Spector[170], Marjo-Riitta Järvelin[168,169,205,206,207,208], Andres Metspalu[146], Christian Kubisch[209], David P. Strachan[210], Michel D. Ferrari[161], Andrea C. Belin[165], Martin Dichgans[163,211], Maija Wessman[145,152], Arn M. J. M. van den Maagdenberg[161,212], John-Anker Zwart[142,143,144], Dorret I. Boomsma[160], George Davey Smith[159], Kari Stefansson[158,213], Nicholas Eriksson[155], Mark J. Daly[138,139,141], Benjamin M. Neale[138,139,141], Jes Olesen[196], Daniel I. Chasman[157,175], Dale R. Nyholt[214] & Aarno Palotie[137,138,139,140,141,145,215]

[137]Psychiatric and Neurodevelopmental Genetics Unit, Massachusetts General Hospital and Harvard Medical School, Boston, MA, USA. [138]Medical and Population Genetics Program, Broad Institute of MIT and Harvard, Cambridge, MA, USA. [139]Stanley Center for Psychiatric Research, Broad Institute of MIT and Harvard, Cambridge, MA, USA. [140]Wellcome Trust Sanger Institute, Wellcome Trust Genome Campus, Hinxton, UK. [141]Analytic and Translational Genetics Unit, Massachusetts General Hospital and Harvard Medical School, Boston, MA, USA. [142]FORMI, Oslo University Hospital, P.O. 4956 Nydalen, 0424 Oslo, Norway. [143]Department of Neurology, Oslo University Hospital, P.O. 4956 Nydalen, 0424 Oslo, Norway. [144]Institute of Clinical Medicine, University of Oslo, P.O. 1171 Blindern, 0318 Oslo, Norway. [145]Institute for Molecular Medicine Finland (FIMM), University of Helsinki, Helsinki, Finland. [146]Estonian Genome Center, University of Tartu, Tartu, Estonia. [147]Division of Endocrinology, Boston Children's Hospital, Boston, MA, USA. [148]Statens Serum Institut, Dept of Epidemiology Research, Copenhagen, Denmark. [149]Novo Nordisk Foundation Center for Basic Metabolic Research, University of Copenhagen, Copenhagen, Denmark. [150]Illumina, 5200 Illumina Way, San Diego, CA, USA. [151]Vall d'Hebron Research Institute, Pediatric Neurology, Barcelona, Spain. [152]Folkhälsan Institute of Genetics, FI-00290 Helsinki, Finland. [153]Neuroscience Center, University of Helsinki, FI-00014 Helsinki, Finland. [154]Research Programs Unit, Molecular Neurology, University of Helsinki, FI-00014 Helsinki, Finland. [155]23andMe, Inc., 899 W. Evelyn Avenue, Mountain View, CA, USA. [156]Inserm Research Center for Epidemiology and Biostatistics (U897), University of Bordeaux, 33076 Bordeaux, France. [157]Division of Preventive Medicine, Brigham and Women's Hospital, Boston, MA 02215, USA. [158]deCODE Genetics, 101 Reykjavik, Iceland. [159]Medical Research Council (MRC) Integrative Epidemiology Unit, University of Bristol, Bristol, UK. [160]VU University Amsterdam, Department of Biological Psychology, 1081 BT Amsterdam, The Netherlands. [161]Leiden University Medical Centre, Department of Neurology, PO Box 9600, 2300 RC Leiden, The Netherlands. [162]Department of

Neurology, Helsinki University Central Hospital, Haartmaninkatu 4, 00290 Helsinki, Finland. [163]Institute for Stroke and Dementia Research, Klinikum der Universität München, Ludwig-Maximilians-Universität München, Feodor-Lynen-Str. 17, 81377 Munich, Germany. [164]Department of Neurology and Epileptology, Hertie Institute for Clincal Brain Research, University of Tuebingen, Tübingen, Germany. [165]Karolinska Institutet, Department of Neuroscience, 171 77 Stockholm, Sweden. [166]Department of Genetics and Computational Biology, QIMR Berghofer Medical Research Institute, 300 Herston Road, Brisbane, QLD 4006, Australia. [167]Ulm University, Institute of Human Genetics, 89081 Ulm, Germany. [168]University of Oulu, Center for Life Course Epidemiology and Systems Medicine, Oulu, Finland. [169]Box 5000, Fin-90014 University of Oulu, Oulu, UK. [170]Department of Twin Research and Genetic Epidemiology, King's College London, London, UK. [171]Department of Epidemiology, Erasmus University Medical Center, 3015 CN Rotterdam, The Netherlands. [172]Department of Radiology, Erasmus University Medical Center, 3015 CN Rotterdam, The Netherlands. [173]Department of Clinical Chemistry, Fimlab Laboratories, and School of Medicine, University of Tampere, Tampere, Finland 33520. [174]Department of Public Health, University of Helsinki, Helsinki, Finland. [175]Harvard Medical School, Boston, MA 02115, USA. [176]University Duisburg Essen, Essen, Germany. [177]Landspitali University Hospital, 101 Reykjavik, Iceland. [178]VU University Medical Centre, Department of Psychiatry, 1081 HL Amsterdam, The Netherlands. [179]Department of Psychiatry, Washington University School of Medicine, 660 South Euclid, CB 8134, St. Louis, MO 63110, USA. [180]University Medical Center Groningen, University of Groningen, 9700RB Groningen, The Netherlands. [181]MRC Functional Genomics Unit, Department of Physiology, Anatomy & Genetics, Oxford University, Oxford, UK. [182]Nuffield Department of Clinical Neuroscience, University of Oxford, Oxford, UK. [183]Oxford Headache Centre, John Radcliffe Hospital, Oxford, UK. [184]Max-Planck-Institute of Psychiatry, Munich, Germany. [185]Christian Albrechts University, Kiel, Germany. [186]Institute of Human Genetics, Helmholtz Center Munich, Neuherberg, Germany. [187]Department of General Practice and Primary Health Care, University of Helsinki and Helsinki University Hospital, Helsinki, Finland. [188]National Institute for Health and Welfare, Helsinki, Finland. [189]Institute of Clinical Medicine, University of Helsinki, Helsinki, Finland. [190]Department of Environmental Health, Harvard T.H. Chan School of Public Health, Boston, MA 02115, USA. [191]Department of Internal Medicine, Erasmus University Medical Center, 3015 CN Rotterdam, The Netherlands. [192]Department of Pain Management and Research, Oslo University Hospital, Oslo, 0424 Oslo, Norway. [193]Medical Faculty, University of Oslo, Oslo, 0318 Oslo, Norway. [194]Division of Mental Health, Norwegian Institute of Public Health,P.O. Box 4404 Nydalen, Oslo NO-0403, Norway. [195]Kiel Pain and Headache Center, 24149 Kiel, Germany. [196]Danish Headache Center, Department of Neurology, Rigshospitalet, Glostrup Hospital, University of Copenhagen, Copenhagen, Denmark. [197]Institute of Biological Psychiatry, Mental Health Center Sct. Hans, University of Copenhagen, Roskilde, Denmark. [198]Institute Of Biological Psychiatry, MHC Sct. Hans, Mental Health Services Copenhagen, DK-2100 Copenhagen, Denmark. [199]Institute of Clinical Sciences, Faculty of Medicine and Health Sciences, University of Copenhagen, DK-2100 Copenhagen, Denmark. [200]iPSYCH—The Lundbeck Foundation's Initiative for Integrative Psychiatric Research, DK-2100 Copenhagen, Denmark. [201]Department of Health, National Institute for Health and Welfare, Helsinki, Finland. [202]Research Centre of Applied and Preventive Cardiovascular Medicine, University of Turku, Turku 20521, Finland. [203]Department of Clinical Physiology and Nuclear Medicine, Turku University Hospital, Turku 20521, Finland. [204]Department of Neurology, Erasmus University Medical Center, 3015 CN Rotterdam, The Netherlands. [205]Imperial College London, Department of Epidemiology and Biostatistics, MRC Health Protection Agency (HPE) Centre for Environment and Health, School of Public Health, London, UK W2 1PG. [206]University of Oulu, Biocenter Oulu, Oulu, Finland. [207]198Box 5000, Fin-90014 University of Oulu, Oulu, UK. [208]Oulu University Hospital, Unit of Primary Care, OuluBox 10, Fin-90029 OYSFinland. [209]University Medical Center Hamburg Eppendorf, Institute of Human Genetics, 20246 Hamburg, Germany. [210]Population Health Research Institute, St George's, University of London, Cranmer Terrace, London SW17 0RE, UK. [211]Munich Cluster for Systems Neurology (SyNergy), Munich, Germany. [212]Leiden University Medical Centre, Department of Human Genetics, PO Box 9600, 2300 RC Leiden, The Netherlands. [213]Faculty of Medicine, University of Iceland, 101 Reykjavik, Iceland. [214]Statistical and Genomic Epidemiology Laboratory, Institute of Health and Biomedical Innovation, Queensland University of Technology, 60 Musk Ave, Kelvin Grove, QLD 4059, Australia. [215]Department of Neurology, Massachusetts General Hospital, Boston, MA, USA.

