## [Peer Review File · Nature Communications]

Reviewers' comments:

Reviewer #2 (Remarks to the Author):

The authors have substantially improved the manuscript. There appears to be some confusion, however, in their use of the terms WMH and SVD. As best as this reviewer can glean, WMH and SVD are synonymous. If this is so, then it might be clearer to stick with WMH since this is the trait that has been analyzed before in the authors' prior studies. If SVD is indeed different from the WMH that the authors' have previously published, then the present manuscript should make that clear. The MEGASTROKE publication to which the authors refer studied what they called "small vessel stroke." Is this also equivalent to SVD and WMH? Perhaps these terms are intuitive to the stroke community. Wider audiences may be tripped up, as this reviewer is.

Reviewer #3 (Remarks to the Author):

The authors perform a large-scale GWAS on white matter hyperintensities on MRI and multiple secondary bioinformatic analyses using including interaction analysis with hypertension, association with related traits, and Mendelian Randomization to test for evidence that WMH (and by proxy small vessel ischemic disease) leads to Alzheimer's and stroke. They perform pathway and cell-type enrichment analyses which are interesting and suggest that genetic variants act via specific cell-types may mediate the risk for small vessel disease. I appreciate that the authors attempt a biological interpretation to the extent it is possible.

Technically, the authors' execution of GWAS and usage of methods seems appropriate. I think the challenges for this study are providing intuitive and coherent interpretations for the dozens of significant but seemingly weak (as far as effect sizes are concerned) associations identified. Additionally, some of the ways by which biological phenomena are quantified and assumed to reflect the underlying pathological process raised concerns for me. These should be addressed so that it is clear to reader what the results actually reflect.

Concerns

- Approach to measuring hypertension as a variable
 - o The authors use criteria of SBP > 140, DBP > 90, or taking antihypertensive medications around the time of MRI as a marker for hypertension.
 - ♣ Can the authors provide some evidence suggesting this approach to classifying patients as hypertensive is a reasonable way of estimating the cumulative effect of hypertension in patients?
 - ♣ It seems like this may be more noise than a true measure of HTN (would explain why the 2-df interaction yields no results, and the aggregate interaction analysis with the 1-df test yields a weakly significant result).
 - ♣ For example, if the authors threshold to uncontrolled SBP > 160, would they see some different effects? Or would there be too few patients to use?
- Approach to measuring white matter hyperintensities
 - o Why take total WMH volume? Would it not be fruitful to quantify it by different brain regions or by specific ROIs supported by either hypotheses or other studies?
 - ♣ E.g. is it possible that juxtacortical WMH and periventricular WMH may have different genetic architecture? Or different pathway and tissue-level associations?
 - ♣ Similarly, is it possible that WMH in specific brain lobes may be more likely to lead to AD?
 - o Have the authors tested whether regional WMH measurements might strengthen some of the associations they observe?
 - ♣ E.g. splitting WMH by different brain regions may be expected to improve ability to identify associations with cell types and eQTLs from brain regions for significant variants.
- Other variables to consider
 - o I appreciate that the authors look at other risk factors including BMI and LDL. What is the role of other factors known to be associated with SVD including BMI, diabetes (Hemoglobin A1c levels),

and smoking?

- WMH genes across the lifespan

o The authors seek to make the inference that the WMH wGRS or individual variants have some effect on WM structure early in life in healthy individuals. I did not find this section very convincing.

o "WMH wGRS showed significant associations with higher MD, RD, and PSMD and lower FA values in i-Share; four WMH risk loci individually showed significant associations with at least one of the DTI parameters"

♣ What does this mean? Is there any longitudinal trend as healthy individuals grow older suggesting a high wGRS leads to a faster decline in WM fiber integrity?

o WMH wGRS is associated with higher PSMD. The authors put a lot of weight on association of PSMD with slower information processing speed on the Stroop test ($P = 0.04$) the poorer performance of those with higher WMH risk scores at younger ages. How did these participants do in other cognitive domains? Was multiple testing accounted for? Does a higher wGRS score also relate to worse performance on the Stroop in this cohort or in other cohorts?

- What is the difference in AD and stroke risk between those with a low vs high WMH wGRS? This would help illustrate to what extent preventing WMH might help reduce risk for these diseases if they are partly mediated by WMH. It would also add interpretability to what are otherwise mostly P values and opaque effect sizes.

Cerebral small vessel disease genomics: implications across the lifespan

Point-by-point response to the reviewers

Reviewers' comments:

Reviewer #2 (Remarks to the Author):

The authors have substantially improved the manuscript. There appears to be some confusion, however, in their use of the terms WMH and SVD. As best as this reviewer can glean, WMH and SVD are synonymous. If this is so, then it might be clearer to stick with WMH since this is the trait that has been analyzed before in the authors' prior studies. If SVD is indeed different from the WMH that the authors' have previously published, then the present manuscript should make that clear. The MEGASTROKE publication to which the authors refer studied what they called "small vessel stroke." Is this also equivalent to SVD and WMH? Perhaps these terms are intuitive to the stroke community. Wider audiences may be tripped up, as this reviewer is.

We thank Reviewer 2 for this comment. WMH and SVD are not synonymous, but as mentioned in the introduction, WMH represent the most common MRI-marker of SVD. MRI-markers of SVD are associated with an increased risk of stroke and dementia, but are distinct from small vessel stroke, which corresponds to a clinical event, with a focal neurological deficit of sudden onset caused by the occlusion of a small artery.

We have clarified the introduction as follows (p.7, line 268-278):

"As a leading cause of stroke, cognitive decline and dementia, cerebral small vessel disease (SVD) represents a major source of morbidity and mortality in aging populations¹⁻⁵. Exploring the mechanisms of SVD and their contribution to dementia risk has recently been identified as a priority research area^{6,7}, based on its more frequent recognition with magnetic resonance imaging (MRI), its high prevalence in older community persons^{4,8} and the demonstration that intensive management of vascular risk factors, especially hypertension, may slow down its progression⁹⁻¹². The biological underpinnings of SVD are poorly understood and no mechanism-based treatments currently are available. White matter hyperintensities of presumed vascular origin (WMH), the most common MRI-marker of SVD, can be measured quantitatively using automated software. They are highly heritable^{13,14}, and confer an increased risk of stroke and dementia⁴, thus making them well-suited to identify potential genetic determinants of SVD and its contribution to stroke and dementia risk. WMH are most often covert, i.e. not associated with a history of clinical stroke. They are highly prevalent in the general population, and much more frequently observed than clinical stroke caused by SVD (which can be both ischemic [small vessel stroke] and hemorrhagic [deep intracerebral hemorrhage]) (Supplementary Fig. 1).

Studying the genomics of SVD also provides a powerful approach to discovery of underlying molecular mechanisms and targets in order to accelerate the development of novel therapies, or identify drug repositioning opportunities¹⁵⁻¹⁷. Although genomic studies of WMH have been most fruitful for deciphering SVD risk variants compared with

other MRI-features of SVD (lacunes, cerebral microbleeds, dilated perivascular spaces)¹⁸ or small vessel stroke¹⁹, or deep intracerebral hemorrhage²⁰, few risk loci have been identified to date²¹⁻²³. This is likely due to limited sample size of populations studied and possibly also the failure to take into account the role of hypertension (HTN), the strongest known risk factor for WMH^{22,23}, in confounding or modifying genetic associations. There is also mounting evidence suggesting that early-life factors play a crucial role in the occurrence of late-life vascular and neurological conditions, including SVD²⁴, likely due to both genetic and environmental factors that may intrinsically influence the vascular substrate of SVD or modulate the brain's resilience to SVD²⁵⁻²⁹. Identifying these early predictors could have major implications for our understanding of disease mechanisms across the lifespan and for devising effective prevention strategies.”

Supplementary Fig. 1

Reviewer #3 (Remarks to the Author):

The authors perform a large-scale GWAS on white matter hyperintensities on MRI and multiple secondary bioinformatic analyses using including interaction analysis with hypertension, association with related traits, and Mendelian Randomization to test for evidence that WMH (and by proxy small vessel ischemic disease) leads to Alzheimer's and stroke. They perform pathway and cell-type enrichment analyses which are interesting and suggest that genetic variants act via specific cell-types may mediate the risk for small vessel disease. I appreciate that the authors attempt a biological interpretation to the extent it is possible.

Technically, the authors' execution of GWAS and usage of methods seems appropriate. I think the challenges for this study are providing intuitive and coherent interpretations for the dozens of significant but seemingly weak (as far as effect sizes are concerned) associations identified. Additionally, some of the ways by which biological phenomena are quantified and assumed to reflect the underlying pathological process raised concerns for me. These should be addressed so that it is clear to reader what the results actually reflect.

Concerns

- Approach to measuring hypertension as a variable
 - o The authors use criteria of SBP > 140, DBP > 90, or taking antihypertensive medications around the time of MRI as a marker for hypertension.
 - ♣ Can the authors provide some evidence suggesting this approach to classifying patients as hypertensive is a reasonable way of estimating the cumulative effect of hypertension in patients?
 - ♣ It seems like this may be more noise than a true measure of HTN (would explain why the 2-df interaction yields no results, and the aggregate interaction analysis with the 1-df test yields a weakly significant result).
 - ♣ For example, if the authors threshold to uncontrolled SBP > 160, would they see some different effects? Or would there be too few patients to use?

We thank Reviewer 2 for these important comments and questions.

- **In the present manuscript, HTN status has been defined following standard guidelines (SBP >140 mmHg, DBP >90 mmHg, or taking antihypertensive medication) (James PA, et al., JAMA 2014). Recently the threshold for defining hypertension has even been lowered to 130/80 mmHg based on the results of the Systolic Blood Pressure Intervention Trial (SPRINT) and lowering blood pressure to the lowest tolerable levels is deemed to yield the greatest clinical benefit (Whelton, J Am Coll Cardiol. 2018; Nasrallah I.M, et al., JAMA 2019; Kjeldsen S.E., et al., Blood Press 2018, Bundy J.D. et al., JAMA Cardiol. 2017). Hence, shifting to a much higher threshold of uncontrolled SBP > 160 mmHg would not reflect this current trend and may not better capture the cumulative effect of HTN.**
- **We have nevertheless now tested the 2-df interaction analysis within the UKBB study (N=19,291) by using two distinct thresholds: 1) with and without HTN defined as SBP >140 mmHg, DBP >90 mmHg, or taking antihypertensive medication (*Classic.HT*, N=9112/10179) and 2) with and without severe uncontrolled hypertension defined as SBP \geq 160mmHG or DBP \geq 100mmHG, with 10 and 5 mmHg added to the SBP and DBP**

levels respectively for participants on antihypertensive medication (*SHT.HT*, $N=2289/17002$). The comparative QQ plot (Figure A below) for these two models shows no substantial difference except for slightly more significant p-values for the top signals with the *SHT.HT* definition, which are however still less significant than in the main effects analysis ($N=19,921$) without any interaction term, thus adding no additional genome-wide significant signals.

Figure A: Quantile-quantile plot comparing the *P*-values from the main-effects and the 2-df interaction test (JMA) with two different definitions for the environmental interaction variable (classic and severe uncontrolled hypertension)

Classic.HT (SBP >140 mmHg, a DBP >90 mmHg, or taking antihypertensive medication)
SHT.HT (SBP ≥ 160 mmHg / DBP ≥ 100 mmHg, with 10 and 5 mmHg added to the SBP and DBP levels for participants on antihypertensive medication)

Moreover, when comparing sub-threshold signals ($5 \times 10^{-8} < P < 5 \times 10^{-6}$) from the JMA analysis with the corresponding main effects we identified 6 independent loci ($r^2 < 0.1$) with a p-value more than 3 orders of magnitude more significant than the joint main genetic effects in the full dataset (CHARGE+UKBB). These may correspond to loci with some GxE interaction with hypertension but insufficient power to reach genome-wide significance in the JMA model. Overall, these 6 loci do not show a more significant JMA result when using *SHT.HT* instead of *Classic.HT* as an environmental interaction variable in the large UKBB subset, rather a trend towards less significant results (Table A below).

Table A: Suggestive associations of genetic loci with White Matter Hyperintensity (WMH) volume at $5 \times 10^{-8} < p < 5 \times 10^{-6}$ in the JMA model, showing more significant p-values than the joint main genetic effects by at least 3 orders of magnitude

Region	Nearest Gene	SNP	Function	EA	EAF	P.value CHARGE+UKBB		P.value UKBB-only	
						Main Effects	JMA(2df)	JMA(2df) Classic-HT	JMA(2df) SHT-HT
1q32.1	SLC41A1	rs6679073	intergenic	c	0.26	4.64E-02	9.12E-07	2.15E-04	3.32E-03
18q12.1	MEP1B	rs673408	intronic	a	0.54	6.31E-01	1.85E-06	2.61E-03	5.67E-01
13q31.3	GPC6	rs9524276	intronic	t	0.78	3.07E-02	3.99E-06	3.59E-01	8.19E-01
3q29	TNK2	rs11720557	intronic	c	0.29	1.81E-01	4.50E-06	3.80E-02	5.26E-01

LOC100133

5q21.1	050	rs114245991	intergenic	g	0.07	8.67E-02	6.12E-06	1.15E-01	5.96E-01
22q13.2	NFAM1	rs73169193	intergenic	c	0.36	2.25E-02	9.94E-06	1.83E-05	4.39E-01

SNP = Single Nucleotide Polymorphism; EA = Effect Allele; EAF = Effect Allele Frequency; JMA = Joint Meta-Analysis; HT = Hypertension; UKBB = UK BioBank

- **We also acknowledge that blood pressure is highly variable and that a one-time blood pressure measurement may not reflect the long-term exposure of participants to high blood pressure levels. Therefore, we had already included a secondary analysis stratifying on quartiles of genetically predicted SBP and DBP levels, which yielded similar results:**
 - o In the results section (p. 8, line 325-327): Associations with WMH volume at these loci were similar in participants with and without HTN and when stratifying on quartiles of genetically predicted SBP and DBP levels (Methods, Supplementary Tables 8-10).
 - o We have also edited the discussion accordingly (p. 14, line 609-612): “We acknowledge limitations. We were underpowered for detecting additional risk variants for WMH after accounting for presence of HTN in the 2-df JMA gene-HTN interaction model. Recognizing that blood pressure is also highly variable and that a one-time blood pressure measurement may not reflect the long-term exposure of participants to high blood pressure levels, we conducted secondary analyses stratifying on quartiles of genetically predicted SBP and DBP levels, yielding similar results. In aggregate, a weighted genetic risk score of independent genome-wide significant WMH risk loci showed a significant 1-df interaction with HTN status in association with WMH volume, suggesting that effect modification of genetic associations by HTN may exist, but that to discover them at the level of individual loci likely will require much larger samples.”

Approach to measuring white matter hyperintensities

Why take total WMH volume? Would it not be fruitful to quantify it by different brain regions or by specific ROIs supported by either hypotheses or other studies? E.g. is it possible that juxtacortical WMH and periventricular WMH may have different genetic architecture? Or different pathway and tissue-level associations?

Similarly, is it possible that WMH in specific brain lobes may be more likely to lead to AD?
o Have the authors tested whether regional WMH measurements might strengthen some of the associations they observe?

E.g. splitting WMH by different brain regions may be expected to improve ability to identify associations with cell types and eQTLs from brain regions for significant variants.

We thank Reviewer 2 for this question. It could indeed be that genetic determinants of WMH volume partly differ according to their location. The most commonly used regional classification is based on the anatomical relationship of WMH to the lateral ventricles in the brain, with a distinction between periventricular white matter hyperintensities (PVWMH) and deep white matter hyperintensities (DWMH). Some epidemiological and pathological data suggest that these may reflect partly distinct underlying mechanisms, with some variations in their prognostic significance. PVWMH and DWMH are however strongly correlated. Moreover, in a parallel effort in the CHARGE Consortium cohorts with such data, represented by a smaller subset of 26,654 participants as the subclassification of WMH is not available in all cohorts, we have carried out GWASs of PVWMH and DWMH.

The manuscript describing the findings is currently in minor revision in another journal. Briefly, using linkage disequilibrium score regression (LDSR), we observed a high genetic correlation between PVWMH and DWMH ($r_g = 0.93$, $p\text{-value} = 1.91E-58$). Using LDSR, we could also demonstrate that both PVWMH and DWMH show a very strong genetic correlation with overall WMH burden derived from the current dataset ($r_g \approx 1$, $p = 1.10E-133$ and $p = 4.28E-35$, respectively). In the PVWMH and DWMH GWAS we identified 11 genome-wide significant loci (11 with PVWMH and 1 with DWMH), of which 7 were previously reported in GWAS of WMH. Two were also identified as genome-wide significant in the present analysis.

To our knowledge there are no genetic studies on WMH volume by brain lobes and there is also no strong rationale for exploring this, as the vascular architecture of small penetrating arteries is not structured by lobes. Other research efforts are underway to explore whether principal component analyses can derive more homogeneous WMH patterns that may better differentiate WMH secondary to lipohyalinosis from WMH secondary to cerebral amyloid angiopathy, but to our knowledge this is not ready for large-scale genetic analyses.

Other variables to consider

I appreciate that the authors look at other risk factors including BMI and LDL. What is the role of other factors known to be associated with SVD including BMI, diabetes (Hemoglobin A1c levels), and smoking?

We thank Reviewer 2 for this helpful suggestion. While the exploration of shared genetic determinants with BMI and diabetes was already included, we have now added an analysis of shared genetic variation with glycated haemoglobin (HbA1c) and lifetime smoking index [a composite measure reflecting the increased exposure to cigarette smoke; capturing smoking heaviness, duration and as well as smoking initiation (SMKindex)], in this revised manuscript, at the genome-wide, regional, and individual variant level. We have also included an exploration of the causal relation between these risk factors and WMH volume. We used summary statistics from the most recent and largest published GWAS of these traits (HbA1c - Wheeler N., et al., Plos Med. 2017; SMKindex – Wooton R.E., et al., Psychol. Med 2019). We have revised the significance threshold for multiple testing accordingly in all analyses. These additional analyses identified important shared genetic variation with smoking. We have updated the main figures (2-4) and added a few additional sentences in the text:

- In the methods section (p. 18) (Line 805-806):
 - “Only SNPs showing an association with a related vascular or neurological trait at $P < 1.3 \times 10^{-4}$ (accounting for 14 independent traits and 27 independent loci) and in moderate to high LD with the lead WMH SNP ($r^2 > 0.50$) are reported. The correlation matrix estimated between the traits using individual-level data from the 3C study was used to estimate the number of independent traits by applying the Matrix Spectral Decomposition (matSpDlite) method.”
- In the results section (pp.9, 10, line 403-428):
 - (p.9, line 403-404) “After correcting for the number of independent loci and traits tested ($P < 1.3 \times 10^{-4}$, Methods), 20 of the 27 WMH risk loci (74%) showed significant association with at least one other trait and/or vascular risk factors.”

- (p.10, line 408-410) “Further significant associations with WMH risk variants were observed for BMI (8 loci), T2D (5 loci), SMKindex (3 loci), and lipid traits (3 loci), one locus (at XKR6) being notably shared with all these risk factors.”
- (p.10, line 417-421) “We observed significant ($P < 3.6 \times 10^{-3}$) genetic correlation of larger WMH volume with higher SBP, DBP, SMKindex, BMI and increased risk of VTE. Using GWAS-PW and HESS (Methods), we identified 16 genomic regions harboring shared genetic risk variants with at least one other vascular trait, predominantly BP traits, but also BMI, lipid levels, and SMKindex (PPA3>0.90).”
- (p.10, line 424-428) “We observed significant ($P < 3.6 \times 10^{-3}$) association of genetically predicted SBP, DBP, PP, SMKindex, and T2D with larger WMH volume and of genetically predicted migraine with smaller WMH volume. After removal of potentially pleiotropic outlier variants the MR-Egger intercept was non-significant for SBP, DBP, PP and SMKindex, indicating no residual pleiotropy and suggesting causal association with WMH volume (Methods).”
- In the discussion section (p. 12, line 536-540):
 - “We additionally show strong causal association between increased exposure to cigarette smoking over the lifetime (lifetime smoking index) and increased WMH burden, as has recently been described in relation with stroke risk⁷⁴, providing some additional evidence for the relevance of smoking cessation to prevent vascular injury and specifically SVD.”

WMH genes across the lifespan

The authors seek to make the inference that the WMH wGRS or individual variants have some effect on WM structure early in life in healthy individuals. I did not find this section very convincing.

“WMH wGRS showed significant associations with higher MD, RD, and PSMD and lower FA values in i-Share; four WMH risk loci individually showed significant associations with at least one of the DTI parameters”

What does this mean? Is there any longitudinal trend as healthy individuals grow older suggesting a high wGRS leads to a faster decline in WM fiber integrity?

WMH wGRS is associated with higher PSMD. The authors put a lot of weight on association of PSMD with slower information processing speed on the Stroop test ($P = 0.04$) the poorer performance of those with higher WMH risk scores at younger ages.

How did these participants do in other cognitive domains?

Was multiple testing accounted for?

Does a higher wGRS score also relate to worse performance on the Stroop in this cohort or in other cohorts?

We thank Reviewer 2 for these comments and questions. We have used five correlated DTI metrics to explore for the first time the association of WMH risk variants with the integrity of the white matter microstructure in young adults (i-Share cohort) and showed a significant association with the same DTI pattern as is observed in cerebral small vessel disease (association of WMH wGRS with higher MD, RD, PSMD and lower FA).

In the i-Share cohort we also observed nominally significant association of increasing PSMD values with slower information processing speed on the Stroop test. This association was not significant after correcting for multiple testing (3 independent DTI markers), given the limited sample size of the i-Share subsample with cognitive testing

(N=1,401), and requires confirmation in additional future samples. It however strongly supports an extension to populations of younger adults (mean age: 22.1 years) of the observation made previously by Baykara et al. regarding association of PSMD with processing speed in cohorts of patients with monogenic and multifactorial SVD and in older population-based samples (mean age ranging from 49.1 – 74.9 years). A more systematic exploration of the clinical significance of PSMD across the lifespan, encompassing various cognitive domains, is currently underway by co-authors on this manuscript, but this is a full project per se that reaches beyond the scope of the present manuscript.

We did not observe any significant association of the WMH wGRS with performance on the Stroop test in i-Share. This is likely related to the small sample size, to the only nominally significant association of PSMD with processing speed, and to the small proportion of PSMD variance explained by the WMH wGRS (0.73%). We observed a trend towards significant association of the WMH wGRS with poorer episodic memory performance (effect estimate \pm SE: -0.19 ± 0.11 , $P=0.08$) using two-sample Mendelian randomization (based on summary statistics from DeBette et al., Biol Psy 2015 in 24,597 older community persons). A more systematic exploration of the clinical significance of the WMH wGRS (and other forthcoming SVD wGRS) across the lifespan, encompassing various cognitive domains, will be conducted as part of independent full projects, beyond the scope of the present manuscript.

We have rephrased the manuscript as follows to introduce more caution in the interpretation of our findings:

- In the results section (pp. 8-9, line 356-360): “Increasing values of PSMD (but not other DTI markers) shows a trend towards association with slower information processing speed on the Stroop test in i-Share participants (N=1,401, effect estimate \pm SE: 0.085 ± 0.040 , $P=0.031$), which did not survive correction for multiple testing (for 3 independent DTI markers). The WMH wGRS was not associated with the Stroop test in i-Share but showed a trend towards an association with poorer episodic memory performance in older community persons (N=24,597, effect estimate \pm SE: -0.19 ± 0.11 , $P=0.08$) (DeBette et al., Biol Psy 2015).”
- In the discussion section (p.13, line 564-591): “Strikingly, our results provide completely novel insight into the lifetime impact of genetic risk for SVD. Indeed, WMH risk variants observed in older adults were already associated with changes in DTI markers of white matter integrity in young adults in their early twenties. Of these, PSMD, a DTI metric recently described to be more strongly correlated with cognitive performance in older persons (patients with sporadic or monogenic SVD and older community persons) than any other MRI-marker of SVD³⁷, was already showing nominal association with lower cognitive performance in young adults. This finding requires confirmation in future independent samples. The association of the WMH wGRS with subtle changes in white matter microstructure in young adults, if confirmed in independent samples, has potential important implications for the timing and paradigm of prediction and prevention of SVD progression and complications. It could reflect that biological pathways contributing to WMH at an older age already have a significant impact on brain microstructure in young adults, possibly reflecting a very early stage of SVD (typically characterized by reduced FA and increased MD and PSMD⁴¹). DTI changes and WMH have been suggested to be dependent physiological processes occurring within

consecutive temporal windows in older patients with SVD^{37,41,85}. Alternatively, observed associations might also reflect pleiotropy between SVD genes and genes influencing brain maturation, as the mean age of i-Share participants corresponds to the peak of white matter maturation⁸⁶. On average FA tends to increase during childhood, adolescence, and early adulthood and then decline in middle-age, while the reverse is observed for MD^{39,40}. Hence the association of the WMH wGRS with lower FA and higher MD could also reflect an impaired or delayed maturation or a premature aging process. The significant association of the WMH wGRS with RD but not A_βD could potentially suggest that this is predominantly reflecting an impact on myelination of fiber tracts⁸⁷, in line with involvement of oligodendroglial dysfunction in early SVD pathology⁸⁸. Future follow-up studies in a longitudinal setting are warranted to better understand the impact of genetically predicted WMH burden on the progression of white matter microstructural changes observed already in young adults and on their link with SVD and its complications.

What is the difference in AD and stroke risk between those with a low vs high WMH wGRS? This would help illustrate to what extent preventing WMH might help reduce risk for these diseases if they are partly mediated by WMH. It would also add interpretability to what are otherwise mostly P values and opaque effect sizes.

We thank Reviewer 2 for this question. The two-sample MR method that we used to describe the causal association of WMH with AD and stroke does not allow checking the difference in their risk between low vs high WMH wGRS. To formally address this question, we would have to conduct analyses at the individual data level to generate quantiles of a weighted WMH wGRS and test the association of extreme quantiles with AD or stroke. Moreover, as has been shown for other common complex diseases, extreme distributions of polygenic risk scores (PRS) for the disease, not restricted to genome-wide significant loci as in the wGRS we have constructed, but with much more liberal significance thresholds, are the most appropriate tool for this type of predictive modeling, and only the top 1% to 8% of the PRS distribution are associated with large odds ratios (>3) to develop the disease compared to the remainder of the population (Khera, Nat Genet 2018). This also requires access to additional very large independent datasets. As part of another ongoing project within the neuroCHARGE consortium and in collaboration with large biobanks, the association a WMH wGRS and PRS (continuous and quantiles) with incident stroke and dementia will be explored, but this is a full project per se and reaches beyond the scope of the present manuscript.

REVIEWERS' COMMENTS:

Reviewer #2 (Remarks to the Author):

None further

Reviewer #3 (Remarks to the Author):

The authors have addressed my concerns about their approaches to and interpretations of their various analyses. I think the paper is very interesting and adds to the growing body of literature better defining both white matter hyperintensities and cerebral small vessel disease.

REVIEWERS' COMMENTS:

Reviewer #2 (Remarks to the Author):

None further

Reviewer #3 (Remarks to the Author):

The authors have addressed my concerns about their approaches to and interpretations of their various analyses. I think the paper is very interesting and adds to the growing body of literature better defining both white matter hyperintensities and cerebral small vessel disease.

➔ We thank the reviewers for their constructive comments and suggestions that have greatly enriched the manuscript.